

# Exploring the Lyapunov instability properties of high-dimensional atmospheric and climate models

Lesley De Cruz[1], Sebastian Schubert[2], Jonathan Demaeyer[1], Valerio Lucarini[2,3,4], and Stéphane Vannitsem[1]

[1]Royal Meteorological Institute of Belgium, Brussels, Belgium
[2]Meteorological Institute, CEN, University Of Hamburg, Germany
[3]Department of Mathematics and Statistics, University of Reading, UK
[4]Centre for the Mathematics of Planet Earth, University of Reading, UK

*Correspondence to:* Lesley De Cruz (lesley.decruz@meteo.be)

**Abstract.** The stability properties of intermediate-order climate models are investigated by computing their Lyapunov exponents (LEs). The two models considered are PUMA (Portable University Model of the Atmosphere), a primitive-equation simple general circulation model, and MAOOAM (Modular Arbitrary-Order Ocean-Atmosphere Model), a quasi-geostrophic coupled ocean-atmosphere model on a $\beta$-plane. We wish to investigate the effect of the different levels of filtering on the

instabilities and dynamics of the atmospheric flows. Moreover, we assess the impact of the oceanic coupling, the dissipation scheme and the resolution on the spectra of LEs.

The PUMA Lyapunov spectrum is computed for two different values of the meridional temperature gradient defining the Newtonian forcing to the temperature field. The increase of the gradient gives rise to a higher baroclinicity and stronger instabilities, corresponding to a larger dimension of the unstable manifold and a larger first LE. The Kaplan-Yorke dimension

of the attractor increases as well. The convergence rate of the rate functional for the large deviation law of the finite-time Lyapunov exponents (FTLEs) is fast for all exponents, which can be interpreted as resulting from the absence of a clear-cut atmospheric time-scale separation in such a model.

The MAOOAM spectra show that the dominant atmospheric instability is correctly represented even at low resolutions. However, the dynamics of the central manifold, which is mostly associated to the ocean dynamics, is not fully resolved because of

its associated long time scales, even at intermediate orders. As expected, increasing the mechanical atmosphere-ocean coupling coefficient or introducing a turbulent diffusion parametrization reduces the Kaplan-Yorke dimension and Kolmogorov-Sinai entropy. In all considered configurations, it is possible to robustly define large deviations laws describing the statistics of the FTLEs corresponding to the strongly damped modes, while the opposite holds for near-zero LEs and, somewhat unexpectedly, also for the positive LEs.

This paper highlights the need to investigate the natural variability of the atmosphere-ocean coupled dynamics by associating rate of growth and decay of perturbations to the physical modes described using the formalism of the covariant Lyapunov vectors and to consider long integrations in order to disentangle the dynamical processes occurring at all time scales.





## 1  Introduction

The dynamics of the atmosphere and the climate system is characterised by the property of sensitivity to initial states (Kalnay, 2003). This feature implies that any small errors in the initial conditions will progressively amplify until the forecast becomes useless, or in other words cannot be distinguished from any random state taken from the climatology of the system. This property was already recognised in the early developments of weather forecasts (Thompson, 1957) and was associated with the nonlinear nature of deterministic dynamical systems by Lorenz (1963). These pioneering works sowed the seeds for the development of predictability theories for the atmosphere and climate, and for important progress in the context of dynamical systems, in particular the development of chaos theory (Eckmann and Ruelle, 1985). This sensitivity property affects not only the dynamics of errors in the initial conditions but also the errors that are present either in the model parametrizations, known as model errors, or in the boundary conditions (Nicolis, 2007; Nicolis et al., 2009). Clarifying the nature of this sensitivity is therefore crucial in the perspective of improving forecasts at short, medium and long term (Vannitsem, 2017)

The property of sensitivity to initial conditions in deterministic dynamical systems is often evaluated by computing the Lyapunov exponents that correspond to the asymptotic rates of amplification or decay of infinitesimally small perturbations, e.g. Eckmann and Ruelle (1985); Ott (2002) and Cencini et al. (2010). A system is chaotic if it possesses at least one positive Lyapunov exponent. Since the eighties many dynamical systems in various domains of science have been analysed from this perspective. This has revealed the presence of chaos in systems ranging from the fields of chemistry and biology to turbulence, e.g. Yamada and Ohkitani (1988); Gallez and Babloyantz (1991); Manneville (1995) and Sprott (2010). In the early days the investigations essentially dealt with low-order systems, but later the scope was broadened to include spatially distributed systems with a large number of degrees of freedom, in coupled maps, e.g. Nicolis et al. (1992); Vannitsem and Nicolis (1996); Cencini et al. (2010) and Yang and Radons (2013), and in partial differential equations, e.g. Manneville (1985, 1995); Vannitsem and Nicolis (1994) and Yang and Radons (2013). Recently, Lyapunov analysis was the subject of a special issue edited by Cencini and Ginelli (2013).

In parallel to these investigations in the context of basic sciences, several attempts to compute Lyapunov exponents in the context of meteorological and climate models have been made, see Vannitsem (2017), in particular in intermediate-order atmospheric quasi-geostrophic models (with O(1000) variables) (Legras and Ghil, 1985; Vannitsem and Nicolis, 1997; Lucarini et al., 2007; Schubert and Lucarini, 2015, 2016). These analyses indicate that if realistic boundary conditions and forcings are imposed on the model under investigation, the number of positive exponents is high, which implies that the solution for the atmosphere lives on a high-dimensional attractor. This suggests at first sight that the number of degrees of freedom needed to describe the dynamics is high and cannot be reduced to a low-order system.

The atmosphere cannot be treated as an autonomous system, as it interacts with other components of the climate system. These other components are characterised by longer time scales of motions. They are typically less intensely affected by some of the physical processes responsible for atmospheric instabilities, and most notably convective and baroclinic instability. Moreover, the energetics of the atmosphere is mainly driven by thermodynamic processes that are dominated by the inhomogeneous absorption of solar radiation. The oceanic circulation, by contrast, is mostly mechanically driven by atmospheric winds



(Lucarini et al., 2014). This raises the question as to the impact of the coupling to other sub-domains of the climate system: are the other sub-domains of the climate system stabilising the atmosphere or not? Vannitsem et al. (2015) partly addressed this question in the context of coupled low-order ocean-atmosphere systems. They found that the presence of the ocean and its exchanges (heat and momentum) with the atmosphere can drastically reduce the instability properties of the flow, confirming earlier results of Nese and Dutton (1993). As discussed below, the role of the ocean in modulating and impacting atmospheric instabilities is far from trivial.

Yet the problem of the predictability (in terms of Lyapunov instability) of the full-scale climate system including the different sub-domains is still open. Recently a new coupled ocean-atmosphere model was developed that could help answer key questions on the predictability properties of this type of system (De Cruz et al., 2016). The model was coined MAOOAM for Modular Arbitrary-Order Ocean-Atmosphere Model. The modular design of MAOOAM allows one to easily explore different model parameters and resolutions. In particular, the coupling strength between the ocean and the atmosphere should modify the predictability properties of the flow as illustrated in (Vannitsem et al., 2015). Moreover, the model resolution is also expected to play an important role in the instability properties of the flow as discussed in (Lucarini et al., 2007) in the context of an atmospheric model.

## 1.1 The properties of the tangent space

As originally envisioned by Ruelle (1979), it is possible to associate to each Lyapunov exponent a corresponding infinitesimal perturbation that co-varies with the orbit that grows or decays asymptotically with the rate given by the corresponding exponent. These physical modes are usually referred to as covariant Lyapunov vectors (CLVs). The application of such a formalism to explore the properties of the tangent space was pioneered by Legras and Vautard (1995) and Trevisan and Pancotti (1998), before Ginelli et al. (2007) and Wolfe and Samelson (2007) provided efficient algorithms to compute them for high-dimensional systems. The CLVs have been used to study e.g. spatio-temporal chaos (Pazó et al., 2008, 2010), Rayleigh-Bénard convection (Xu and Paul, 2016), and the dynamics of the mid-latitudes atmosphere in the quasi-geostrophic (QG) approximation (Schubert and Lucarini, 2015, 2016). Schubert and Lucarini (2015, 2016) have also underlined that CLVs allow for generalising the classic normal mode instability of fixed stationary states to the case of chaotic background state (e.g. Charney, 1947; Eady, 1949; Pedlosky, 1964). Trevisan et al. (2010) and Carrassi et al. (2008) also showed that performing data assimilation on the unstable manifold spanned by the CLVs corresponding to positive Lyapunov exponents is extremely efficient because it allows one to focus on the portion of the tangent space supporting the growth of errors.

Additionally, CLVs allow for understanding the properties of the tangent space and assess the hyperbolicity of the system, through the analysis of the statistics of the angles between the stable and unstable tangent manifolds across the attractor. These angles should always be bounded away from 0 or $\pi$ in the ideal case of uniform hyperbolicity. This point of view complements the investigation of the statistical properties of FTLEs: the probability density functions of the FTLEs whose long term averages correspond to positive exponents do not cross zero in the case of uniform hyperbolicity. Note that the uniform hyperbolicity is key to defining the structural stability of a chaotic dynamical system and provides, through the chaotic hypothesis by Gallavotti and Cohen (1995), an important working hypothesis for constructing the statistical mechanics of high-dimensional chaotic



systems, *even in the case that such a system is not uniformly hyperbolic*. Note that uniform hyperbolicity also allows for establishing a rigorous response theory for chaotic dynamical systems (Ruelle, 2009), which has also been shown to apply well in complex systems where there is no reason to believe that such stringent condition on the tangent space is obeyed; see, e.g., Lucarini et al. (2017).

## 1.2   Multiscale Properties

As well known, geophysical fluid dynamical (GFD) systems are characterised by relevant processes on multiple spatial and temporal scales of motion (Schneider, 2006; Vallis, 2006). These scales of motion can be isolated by assuming dominant dynamical balances and performing corresponding asymptotic expansions of the dynamical equations (Klein, 2010). A possible way to look at the signature of such a diverse range of dynamical processes in a nonlinear, chaotic setting can be found by considering the general idea proposed by Gallavotti and Lucarini (2014) according to which one can expect to find that LEs corresponding to smaller time scales are associated to CLVs characterised by small spatial scales. By looking at the properties of the structure of each CLV, one should ideally be able to understand what kind of dynamical processes (e.g. QG vs. mesoscale) are mainly responsible for such a physical mode.

The problem becomes particularly interesting when considering the coupling of two sub-domains with vastly different time scales as done in the case of a low-order coupled ocean-atmosphere system in Vannitsem and Lucarini (2016). Three different manifolds were isolated in the model, the usual (highly) unstable manifold mainly associated with the dynamics of the atmosphere, a highly dissipative manifold also mainly associated with the dynamics of the atmosphere, and an extremely weakly (un-) stable manifold, that will be here referred to as the *central manifold*, essentially dominated by the dynamics and thermodynamics of the ocean but coupled to the atmosphere as well. The presence of a nontrivial central manifold is typical of the so-called partially hyperbolic systems (Pesin, 2004). The CLVs corresponding to the central manifold are geometrically quasi-degenerate, so that errors propagate easily between the various modes and impact both the atmosphere and the ocean. The corresponding FTLEs are strongly correlated and each have a rather slow decay of correlations, so that large deviation laws cannot be effectively estimated (Kifer, 1990; Touchette, 2009; Pazó et al., 2013; Laffargue et al., 2013). A particular consequence of this feature is that errors affecting the central manifold display a complex super-exponential behaviour. The question is therefore what should be the resolution of the coupled atmosphere-ocean model and what should be the time of observation such that a better separation emerges between such modes.

## 1.3   This paper

In this paper we wish to provide some first steps of a wider research programme aimed at performing a systematic investigation of the properties of the tangent space of GFD systems in a turbulent regime of motion. A first objective is to gain a better understanding of the multiscale properties of the dynamics and of the energy exchanges occurring across such scales. Furthermore, this programme aims at understanding the relevance of violations to the uniform hyperbolicity conditions in terms of predictability on different time scales, including the response – in a statistical mechanical sense – of the system to static and time-dependent perturbations.



In the present manuscript, we explore for the first time the Lyapunov spectra of a primitive-equation model, PUMA, and of the intermediate-order coupled ocean-atmosphere system, MAOOAM. The first model is characterised by the presence of multiple scales of motions resulting from the fact that ageostrophic motions are *not* filtered, as opposed to the QG case (Klein, 2010). Instead, in the second model the multiscale properties come from the fact that the represented two geophysical fluids have largely different internal time scales.

For PUMA, we consider the first 200 Lyapunov exponents for two different meridional temperature gradients. We study the properties of the Lyapunov spectrum and on the estimates of the Kaplan-Yorke dimension and Kolmogorov-Sinai entropy (Eckmann and Ruelle, 1985). In the case of MAOOAM, we investigate the role of dissipation introduced in the model (linear friction and effective diffusion) and the impact of the resolution of the models. For both models, the existence of large deviation laws of the FTLEs is tested.

In Section 2, the two models are described and Section 3 is devoted to a brief description of the Lyapunov instability analysis and the experimental setups. Section 4 summarises the results obtained so far and in Section 5 we present our future programme, which aims to clarify the instabilities of high-resolution systems.

## 2   Model description

### 2.1   PUMA

The Portable University Model of the Atmosphere (PUMA) was introduced by Fraedrich et al. (1998). The intent of its developers was to design a model that gets close to state-of-the-art circulation models and at the same time is still easy to use in teaching and research by single scientists. PUMA is the dynamical core of the Planet Simulator (PLASIM) which is a fully coupled climate model of intermediate complexity. PLASIM has been frequently used to study storm tracks (Fraedrich and Kirk, 2005), explore the sensitivity of the sea ice albedo bifurcation (Boschi et al., 2013) or create large ensembles for climate change experiments allowing for interesting applications in economic modelling (Holden et al., 2013) or to assess the feasibility of linear response theory (Ragone et al., 2015).

Let us briefly summarise the equations of motion of PUMA and how the model is integrated in time. For further details we refer the reader to the PUMA User's Guide (Fraedrich et al., 1998). PUMA solves the primitive equations, which are derived from the Navier-Stokes equation on a rotating sphere by assuming approximate hydrostatic balance. This means that (convective) motions which are characterised by a vertical acceleration with a size comparable to gravity are filtered out (Holton, 2004). The prognostic equations as written in PUMA's code have four prognostic fields, the relative vorticity $\eta$, the



divergence $D$, the temperature $T$ and the logarithm of the surface pressure $\ln p_s$. These equations are

$$\frac{\partial (\eta + f)}{\partial t} = \frac{1}{1 - \mu^2} \frac{\partial F_v}{\partial \lambda} - \frac{\partial F_u}{\partial \mu} + P_\eta \tag{1}$$

$$\frac{\partial D}{\partial t} = \frac{1}{1 - \mu^2} \frac{\partial F_u}{\partial \lambda} - \frac{\partial F_v}{\partial \mu} - \left( \frac{U^2 + V^2}{2(1 - \mu^2)} + \Phi + T_0 \ln p_s \right) + P_D \tag{2}$$

$$\frac{\partial p_s}{\partial t} = - \int_0^1 A d\sigma \tag{3}$$

$$\frac{\partial T'}{\partial t} = -\frac{1}{1 - \mu^2} \frac{\partial (UT')}{\partial \lambda} - \frac{\partial (VT')}{\partial \mu} + DT' - \dot{\sigma} \frac{\partial T}{\partial \sigma} + \kappa \frac{T}{p} \omega + \frac{J}{c_p} + P_T \tag{4}$$

where the vorticity is defined as $\eta = \partial_x v - \partial_y u$ and the divergence is defined as $D = \partial_x u + \partial_y v$. Additionally, one takes into account the hydrostatic relation

$$\frac{\partial \Phi}{\partial \ln \sigma} = -T \tag{5}$$

and $T'$ is defined as $T' = T - T_0$ with $T_0 = 250K$. Some abbreviations have been used:

$$F_u = V(\eta + f) - \dot{\sigma} \frac{\partial U}{\partial \sigma} - T' \frac{\partial \ln p_s}{\partial \lambda}$$

$$F_v = -U(\eta + f) - \dot{\sigma} \frac{\partial V}{\partial \sigma} - T'(1 - \mu^2) \frac{\partial \ln p_s}{\partial \mu}$$

$$A = D + \boldsymbol{V} \cdot \nabla \ln p_s$$

with $U = u \cos \phi$ and $V = v \cos \phi$. The variables used in this equation can be found in Table 1.

PUMA is forced by Newtonian cooling which accounts in a crude yet effective way for the emission and the absorption of long and short wave radiation and for the heat convergence associated to convective processes (following Held and Suarez (1994)). This process is described by the equations

$$\frac{J}{c_p} + P_T = \frac{T_R(\phi, \sigma) - T}{\tau_R} + H_T \tag{6}$$

$$T_R(\phi, \sigma) = T_R^{vert}(\sigma) + f(\sigma) T_R^{hor}(\phi), \tag{7}$$

where $T_R$ is the temperature restoration field that depends on the fixed meridional pole-to-equator temperature gradient $\Delta T_{EP}$ and the pole-to-pole gradient $\Delta T_{NS}$. The latter gradient is zero in our experiments, so that we have equatorial symmetry in our boundary conditions and each solution we find is accompanied by a mirrored solution at the equator. For completeness, we





also add the full equations of the restoration field, and we refer the reader to Fraedrich et al. (1998) for a more detailed account:

$$T_R^{hor}(\phi) = \Delta T_{NS} \frac{\sin(\phi)}{2} - \Delta T_{EP} \left( \sin^2 \phi - \frac{1}{3} \right) \tag{8}$$

$$T_R^{vert}(\sigma) = \Delta T_{trop} + \sqrt{\frac{L}{2} \left( z_{tp} - z(\sigma) \right)^2 + S^2} + \frac{L}{2} \left( z_{tp} - z(\sigma) \right) \tag{9}$$

$$f(\sigma) = \begin{cases} \sin \left( \frac{\pi}{2} \left( \frac{\sigma - \sigma_{tp}}{1 - \sigma_{tp}} \right) \right) & \text{, if } \sigma \geq \sigma_{tp} \\ 0 & \text{, if } \sigma < \sigma_{tp}. \end{cases} \tag{10}$$

Here, $\sigma_{tp}$ is the height of the tropopause, whereas $z_{tp}$ is the global constant height of the tropopause. $S$ is a technical smoothing parameter. Finally, the hyperdiffusion $H_T$ in Eq. (6) is defined as $H_T = \nabla^8 T$ and parametrizes small scale interactions.

PUMA uses spherical harmonics and grid point fields of the prognostic variables. Utilising the Fourier transform along the zonal direction and a Legendre transformation, PUMA computes the linear terms in spectral space and the nonlinear terms in grid point space. The time-stepping scheme is a combination of a leap-frog scheme with Robert-Asselin filter.

The PUMA User's Guide includes more details and a complete description of the exact implementation and form of the various forcings (Fraedrich et al., 1998).

## 2.2 MAOOAM

Although the atmospheric dynamics of both models are largely governed by the same processes, MAOOAM differs in many respects from the stand-alone PUMA model. Most importantly, the atmosphere of MAOOAM features both a mechanical and a
thermal coupling with a shallow-water ocean layer, which is absent in PUMA. Furthermore, MAOOAM is a mid-latitude model which uses the quasi-geostrophic approximation (Charney and Straus, 1980) on a $\beta$-plane (Vallis, 2006), whereas PUMA is a global primitive-equation model, in which the filtering is applied at a much smaller scale. The representation of the dynamical fields differs accordingly, with MAOOAM adopting a Fourier basis, using products of sine and cosine functions that respect the boundary conditions of a zonally periodic atmosphere over a rectangular ocean basin (De Cruz et al., 2016).

The dynamics of MAOOAM's two-layer atmosphere is described by the quasi-geostrophic vorticity equations, expressed in terms of the streamfunction fields $\psi_a^1$ at 250 hPa and $\psi_a^3$ at 750 hPa as in Charney and Straus (1980),

$$\frac{\partial}{\partial t} \left( \nabla^2 \psi_a^1 \right) + J(\psi_a^1, \nabla^2 \psi_a^1) + \beta \frac{\partial \psi_a^1}{\partial x} = -k_d' \nabla^2 (\psi_a^1 - \psi_a^3) + \frac{f_0}{\Delta p} \omega, \tag{11}$$

$$\frac{\partial}{\partial t} \left( \nabla^2 \psi_a^3 \right) + J(\psi_a^3, \nabla^2 \psi_a^3) + \beta \frac{\partial \psi_a^3}{\partial x} = +k_d' \nabla^2 (\psi_a^1 - \psi_a^3) - \frac{f_0}{\Delta p} \omega - k_d \nabla^2 (\psi_a^3 - \psi_o), \tag{12}$$

in which the vertical velocity $\omega$ can be eliminated by applying the hydrostatic relation and the ideal gas law, as detailed in
(De Cruz et al., 2016).

Following Pierini (2011), the equation of motion for the ocean layer is described by

$$\frac{\partial}{\partial t} \left( \nabla^2 \psi_o - \frac{\psi_o}{L_R^2} \right) + J(\psi_o, \nabla^2 \psi_o) + \beta \frac{\partial \psi_o}{\partial x} = -r \nabla^2 \psi_o + \frac{C}{\rho h} \nabla^2 (\psi_a^3 - \psi_o). \tag{13}$$



**Table 1.** Symbols and variables in the PUMA equations.

| Symbol | Description |
|---|---|
| $T$ | temperature |
| $T_0$ | reference temperature |
| $T' = T - T_0$ | temperature deviation from $T_0$ |
| $\eta$ | relative vorticity |
| $D$ | divergence |
| $p_s$ | surface pressure pressure |
| $\Phi$ | geopotential |
| $t$ | time |
| $\lambda, \phi$ | longitude, latitude |
| $\mu = \sin\phi$ | |
| $\sigma = \frac{p}{p_s}$ | sigma vertical coordinate |
| $\dot{\sigma} = \frac{d\sigma}{dt}$ | vertical velocity in $\sigma$-system |
| $\dot{p} = \frac{dp}{dt}$ | vertical velocity in $p$-system |
| $u, v$ | zonal, meridional component of horizontal velocity |
| $\boldsymbol{V}$ | horizontal velocity with components $U, V$ |
| $f$ | Coriolis parameter |
| $J$ | diabatic heating rate |
| $c_p$ | specific heat of dry air at constant pressure |
| $\kappa$ | adiabatic coefficient |

The prognostic equations for the atmospheric and oceanic temperature fields, using an energy balance scheme as in Barsugli and Battisti (1998), are

$$\gamma_{\mathrm{a}} \left( \frac{\partial T_{\mathrm{a}}}{\partial t} + J(\psi_{\mathrm{a}}, T_{\mathrm{a}}) - \sigma\omega\frac{p}{R} \right) = -\lambda(T_{\mathrm{a}} - T_{\mathrm{o}}) + \epsilon_{\mathrm{a}}\sigma_{\mathrm{B}}T_{\mathrm{o}}^4 - 2\epsilon_{\mathrm{a}}\sigma_{\mathrm{B}}T_{\mathrm{a}}^4 + R_{\mathrm{a}}, \tag{14}$$

$$\gamma_{\mathrm{o}} \left( \frac{\partial T_{\mathrm{o}}}{\partial t} + J(\psi_{\mathrm{o}}, T_{\mathrm{o}}) \right) = -\lambda(T_{\mathrm{o}} - T_{\mathrm{a}}) - \sigma_{\mathrm{B}}T_{\mathrm{o}}^4 + \epsilon_{\mathrm{a}}\sigma_{\mathrm{B}}T_{\mathrm{a}}^4 + R_{\mathrm{o}}. \tag{15}$$

The quartic terms in these equations are linearised by decomposing the temperature fields around a spatially and temporally constant equilibrium temperature, $T_{\mathrm{a}} = T_{\mathrm{a}}^0 + \delta T_{\mathrm{a}}$ and $T_{\mathrm{o}} = T_{\mathrm{o}}^0 + \delta T_{\mathrm{o}}$, and solving the quartic equation for the equilibrium temperature (Vannitsem et al., 2015).

The thermal wind relation allows one to link the atmospheric temperature anomaly $\delta T_{\mathrm{a}}$ to the baroclinic streamfunction $\theta_{\mathrm{a}} \equiv (\psi_{\mathrm{a}}^1 - \psi_{\mathrm{a}}^3)/2$, more specifically $\delta T_{\mathrm{a}} = 2f_0\theta_{\mathrm{a}}/R$. Hence, the remaining independent dynamical fields are the barotropic atmospheric streamfunction field $\psi_{\mathrm{a}}$, defined as $\psi_{\mathrm{a}} \equiv (\psi_{\mathrm{a}}^1 + \psi_{\mathrm{a}}^3)/2$, the oceanic streamfunction field $\psi_{\mathrm{o}}$, and the temperature anomalies $\delta T_{\mathrm{a}}$ and $\delta T_o$ of the atmosphere and the ocean. The other parameters and variables that feature in the MAOOAM model equations are explained in Table 2.



**Table 2.** Variables and parameters in the MAOOAM equations

| Variable (units) | Description |
| --- | --- |
| $\psi_\mathrm{o}, \psi_\mathrm{a}$ (m$^2$ s$^{-1}$) | streamfunction of the ocean, atmosphere |
| $\omega = dp/dt$ (Pa s$^{-1}$) | vertical velocity in pressure coordinates |
| $T_\mathrm{o}, T_\mathrm{a}$ (K) | temperature of the ocean, atmosphere; $T_x = T_x^0 + \delta T_x$ |
| $\delta T_\mathrm{o}, \delta T_\mathrm{a}$ (K) | temperature anomaly of the ocean, atmosphere |

| Parameter (units) | Description |
| --- | --- |
| $n = 2L_y/L_x$ | meridional to zonal aspect ratio |
| $L_y = \pi L$ (km) | meridional extent of the domain |
| $f_0$ (s$^{-1}$) | Coriolis parameter at 45° latitude |
| $\lambda$ (W m$^{-2}$ K$^{-1}$) | heat transfer coefficient at the ocean-atmosphere interface |
| $r$ (s$^{-1}$) | friction coefficient at the bottom of the ocean layer |
| $C_\mathrm{o}, C_\mathrm{a}$ (W m$^{-2}$) | insolation coefficient of the ocean, atmosphere |
| $k_d$ (s$^{-1}$) | friction coefficient at ocean-atmosphere interface |
| $k_d'$ (s$^{-1}$) | friction coefficient between the atmospheric layers |
| $h$ (m) | depth of the ocean layer |
| $d = C/(\rho h)$ (s$^{-1}$) | mechanical ocean-atmosphere coupling coefficient |
| $R$ (J kg$^{-1}$ K$^{-1}$) | gas constant of dry air |
| $L_\mathrm{R}$ (km) | reduced Rossby deformation radius of the ocean |
| $\rho$ (kg m$^{-3}$) | density of the ocean |
| $\sigma_\mathrm{B}$ (W m$^2$ K$^{-4}$) | Stefan-Boltzmann constant |
| $\sigma$ (m$^2$ s$^{-2}$ Pa$^{-2}$) | static stability of the atmosphere |
| $\beta$ (m$^{-1}$ s$^{-1}$) | Rossby parameter $\frac{df}{dy}$ |
| $\gamma_\mathrm{o}, \gamma_\mathrm{a}$ (J m$^{-2}$ K$^{-1}$) | Specific heat capacity of the ocean layer, atmosphere |
| $T_\mathrm{o}^0, T_\mathrm{a}^0$ (K) | constant solution for the temperature of the ocean, atmosphere |
| $\epsilon_\mathrm{a}$ | grey-body atmosphere emissivity |

The model equations are nondimensionalized, and the dynamical fields are expanded in a configurable set of Fourier modes. The MAOOAM code computes the coefficients for the resulting set of ordinary differential equations (ODEs) as algebraic formulae of the wavenumbers. These ODEs are then integrated using a fourth-order Runge-Kutta integration scheme. We refer the reader to De Cruz et al. (2016) for more details on the expansion of the dynamical fields in terms of Fourier modes, the

5   computation of the coefficients, and the tensorial implementation of the ODEs.

In what follows, we will use a shorthand notation that uses the maximum wavenumbers $N_x$ and $N_y$ to specify the model resolution. If the resolution of the ocean and the atmosphere are the same, the model resolution is referred to as $N_x$x$N_y$; otherwise, it is denoted as atm. $N_{x,\mathrm{a}}$x$N_{y,\mathrm{a}}$, oc. $N_{x,\mathrm{o}}$x$N_{y,\mathrm{o}}$.





## 3  Methodology

### 3.1  Computation of the Lyapunov exponents

Let us write the evolution laws of the autonomous system presented in Section 2 in a synthetic form

$$\frac{d\boldsymbol{x}}{dt} = \boldsymbol{f}(\boldsymbol{x}, \alpha) \tag{16}$$

5  where $\boldsymbol{x}$ is a vector containing the entire set of relevant variables $\boldsymbol{x} = (x_1, ..., x_N)$ such as temperature or wind velocity, projected on a relevant set of modes as described in Section 2. The function $\boldsymbol{f}$ is a nonlinear function of the variables $\boldsymbol{x}$ and $\alpha$ represents a set of parameters.

Let us consider a small perturbation along the trajectory, $\boldsymbol{x}(t)$, generated by model (16), denoted $\delta\boldsymbol{x}(t)$. Provided that this perturbation is sufficiently small (ideally infinitely small), its dynamics can be described by the linearised equation,

$$\frac{d\delta\boldsymbol{x}}{dt} = \left.\frac{\partial \boldsymbol{f}}{\partial \boldsymbol{x}}\right|_{\boldsymbol{x}(t)} \delta\boldsymbol{x} \tag{17}$$

and a formal solution can be written as

$$\delta\boldsymbol{x}(t) = \mathbf{M}(t, t_0)\delta\boldsymbol{x}(t_0) \tag{18}$$

where the matrix $\mathbf{M}$ is referred to as the resolvent matrix. This matrix $\mathbf{M}$ is responsible for the amplification or contraction of the errors during the time period $t - t_0$. In order to get information independent of the initial or final time, a limit $(t - t_0) \to \infty$

15  should be taken. Oseledets (1968, 2008) demonstrates that provided that the system is ergodic, the following limit exists for almost all initial conditions $\boldsymbol{x}(t_0) = \boldsymbol{x}_0$,

$$\lim_{t \to \infty} (M^* M)^{1/2(t - t_0)} = \Lambda_{x_0} \tag{19}$$

The *Lyapunov exponents* are then defined as the natural logarithm of the eigenvalues of $\Lambda_{x_0}$. These are usually represented in decreasing order and the full set of exponents is called the *Lyapunov spectrum*. Other definitions are available but will not be

20  discussed here since we do not use them in this study. These can be found in (Eckmann and Ruelle, 1985; Legras and Vautard, 1995), and in a recent work in the context of the coupled ocean-atmosphere (Vannitsem and Lucarini, 2016).

The computation of the backward Lyapunov exponents follows the standard algorithm of (Shimada and Nagashima, 1979; Benettin et al., 1980) based on the Gram-Schmidt orthogonalisation.

1. An ensemble $E$ of $N$ perturbation vectors is randomly initialised.

2. Every time step, the model propagator is computed from the tangent linear model. This is the matrix that quantifies the transition from one model state into that one time step later.

3. The model is integrated forward in time, and the propagators are accumulated (multiplied) into a matrix $P$.





4. Every $b$ time steps, $E$ is evolved with $P$, and Gram-Schmidt orthogonalised (using a $QR$-decomposition). The local Lyapunov spectrum is computed from the diagonal of $R$.

5. Mean and variance of the local Lyapunov exponents are calculated.

The full Lyapunov spectrum of a model allows us to compute some additional interesting properties of its attractor. One of these is the Kaplan-Yorke or Lyapunov dimension $D_{KY}$, which provides (a lower bound on) the fractal dimension of the attractor, and is defined as (Frederickson et al., 1983):

$$D_{KY} = k + \frac{\lambda_1 + \lambda_2 + \ldots + \lambda_k}{|\lambda_{k+1}|}, \tag{20}$$

where $k$ is the highest index for which the sum of the largest $k$ Lyapunov exponents is still strictly positive.

The second important property of the attractor is the Kolmogorov-Sinai or metric entropy $h_{KS}$, a quantity that describes the rate of growth of the Shannon entropy (Eckmann and Ruelle, 1985; Boffetta et al., 2002), which characterises the quantity of information necessary to locate the solution on its attractor. Its upper bound is the sum of the positive Lyapunov exponents:

$$h_{KS} \leq \sum_{\lambda_i > 0} \lambda_i. \tag{21}$$

with the equality proven for a very particular class of systems, known as Axiom A systems. Here the KS entropy will be assumed to be close to the sum of positive exponents, and hence this sum will be referred to as the KS entropy.

## 3.2 Large deviation laws

Since the Lyapunov exponents are obtained by considering limiting conditions where the initial perturbations are very small and the time span over which the growth or decay rate is very long, they cannot reasonably be used to study predictability outside such conditions. Finite-time Lyapunov exponents (FTLEs) (e.g. Haller, 2000) have been proposed to address such shortcomings, with the caveat that they do not enjoy the extremely beneficial mathematical properties (especially, norm-independence) that characterise the Lyapunov exponents.

In this paper, we focus on the FTLEs and their relation to the asymptotic mean LEs. Hence, we are interested in averages $\sigma_M^j$ over a time $M$ of one backward Lyapunov exponent $\lambda_j$ and its statistics. If a dynamical system is an Axiom A system or – invoking the chaotic hypothesis – one of a certain type of non Axiom A systems, these fluctuations for a finite, but large $M$ may be described (based on (Schalge et al., 2013; Pazó et al., 2013; Laffargue et al., 2013)) by a large deviation law (Kifer, 1990; Touchette, 2009). For the finite-time backward LEs and for a large $M$, we will verify the following relation for the distribution $\mathcal{P}$ of the averages:

$$\mathcal{P}\left(\sigma_M^j = x\right) \propto \exp\left(-M\,I_j(x)\right). \tag{22}$$

$I_j(x)$ is the rate function, which is independent of $M$. The rate function can be computed directly from this relation as $I_j(x) = \lim_{M \to \infty} -\frac{1}{M} \log\left(\mathcal{P}\left(\sigma_M^j = x\right)\right)$.





If $x$ represents a time series, we have to take the autocorrelation into account. The FTLEs for the models under consideration have a non-zero autocorrelation. To account for this, the time series are decomposed into blocks that are decorrelated. For each LE, we find the smallest block size, called the decorrelation time $T_{\text{decorr}}$. The time $T_{\text{decorr}}$ is chosen to be the time lag when the autocorrelation drops below $1/e$.

## 3.3 Experimental design: PUMA

We choose a simple setup of PUMA. In this spirit, we also switch off orography. The system is forced via a constant temperature gradient between the equator and the respective poles, as detailed in Section 2.1. We conduct simulations at a horizontal resolution of T42, which amounts to roughly 250 km. In grid-point space this corresponds to a Gaussian grid with 64 latitudes and 128 longitudes. In the vertical direction we restrict the resolution to 10 sigma levels. The integration scheme uses a time step of one hour.

The objective of our experiments with PUMA is to compute the backward Lyapunov exponents. For this we perform spin-up simulations for 30 years from random initial conditions. We then obtain the first 200 Lyapunov exponents using the Benettin algorithm described in Section 3.1. We allow the algorithm to converge for 5 years and finally obtain a time series of 25 years for all LEs. In order to explore two different chaotic regimes with many positive LEs, we perform two experiments with an equator-to-pole temperature gradient $T_{EP}$ of 50 K and 60 K, respectively (with $\Delta T_{NS} = 0$).

Note that in order to compute the Lyapunov exponents, it is necessary to construct the tangent linear of PUMA. We generated parts of the code using the program TAF by *FastOpt* (Giering and Kaminski, 2003).

## 3.4 Experimental design: MAOOAM

Table 3 lists the values of the physical parameters that are used in the present study. These are selected to lie within the realistic ranges previously derived by Vannitsem and De Cruz (2014), and correspond to the setup used by De Cruz et al. (2016). In addition, we explore different values of the mechanical ocean-atmosphere coupling coefficient $d$ and the eddy viscosity coefficients $\nu_{\text{a}}$ and $\nu_{\text{o}}$, as well as a range of model resolutions.

All experiments are performed with the same integration parameters. The time step of 0.2 nondimensional time units corresponds to 32.3 minutes in dimensional units. Before calculating the Lyapunov spectrum, a transient run of $10^8$ nondimensional time units is performed, corresponding to 30726 years. Using the tangent linear model of MAOOAM, the backward Lyapunov exponents are then computed using the algorithm described in Section 3.1. In our simulations, the orthogonalisation is performed every time step, i.e. $b = 1$. The Lyapunov spectrum is computed from simulations of 614 years.

The experiments are performed for different resolutions as discussed in Section 2 and for different dissipation schemes as described below.

– nodissip

This experiment corresponds to the setup of De Cruz et al. (2016). In addition to the variables listed in Table 3, the mechanical ocean-atmosphere coupling parameter $d$ is set to $1.1 \times 10^{-7}$ s$^{-1}$.





**Table 3.** Common parameter values for the different model configurations of MAOOAM.

| Parameter (unit) | Value | Parameter (unit) | Value |
|---|---|---|---|
| $n = 2L_y/L_x$ | 1.5 | $L_R$ (km) | 19.93 |
| $L_y = \pi L$ (km) | $5.0 \times 10^3$ | $\rho$ (kg m$^{-3}$) | 1000 |
| $f_0$ (s$^{-1}$) | $1.032 \times 10^{-3}$ | $\sigma_B$ (W m$^2$ K$^{-4}$) | $5.6 \times 5.610^{-8}$ |
| $\lambda$ (W m$^{-2}$ K$^{-1}$) | 15.06 | $\sigma$ (m$^2$ s$^{-2}$ Pa$^{-2}$) | $2.16 \times 10^{-6}$ |
| $r$ (s$^{-1}$) | $1.0 \times 10^{-7}$ | $\beta$ (m$^{-1}$ s$^{-1}$) | $1.62 \times 10^{-11}$ |
| $C_o$ (W m$^{-2}$) | 310 | $\gamma_o$ (J m$^{-2}$ K$^{-1}$) | $5.46 \times 10^8$ |
| $C_a$ (W m$^{-2}$) | $C_o/3$ | $\gamma_a$ (J m$^{-2}$ K$^{-1}$) | $1.0 \times 10^7$ |
| $k_d$ (s$^{-1}$) | $3.0 \times 10^{-6}$ | $T_a^0$ (K) | 289.30 |
| $k_d'$ (s$^{-1}$) | $3.0 \times 10^{-6}$ | $T_o^0$ (K) | 301.46 |
| $h$ (m) | 136.5 | $\epsilon_a$ | 0.7 |
| $R$ (J kg$^{-1}$ K$^{-1}$) | 287 | | |

– nodissip-reducedstress

For this "reduced-stress" experiment, the coupling parameter $d$ is reduced to $4 \times 10^{-8}$ s$^{-1}$.

– dissipation

One of the physical processes that was not included in MAOOAM v1.0 (De Cruz et al., 2016) is the kinematic dissipa-
tion of energy due to turbulent diffusion, which becomes increasingly important at smaller spatial scales. This process is
parametrized as a dissipation term in the prognostic equations for the atmospheric (barotropic) and oceanic streamfunc-
tion, that is proportional to the squared Laplacian of the respective streamfunction:

$$D_o = \nu_o \nabla^4 \psi_o, \tag{23}$$
$$D_a = \nu_a \nabla^4 \psi_a. \tag{24}$$

We adopt the values for the parameters $\nu_o$ and $\nu_a$ from Van der Avoird et al. (2002), where they are estimated to be

$$\nu_o = 1.76 \times 10^4 \text{ m}^2 \text{ s}^{-1}, \tag{25}$$
$$\nu_a = 1 \times 10^5 \text{ m}^2 \text{ s}^{-1}. \tag{26}$$

Furthermore, $d$ is set to $1.1 \times 10^{-7}$ $s^{-1}$.

– dissipationx10

In this experiment, $d = 1.1 \times 10^{-7}$ $s^{-1}$, but the parameters $\nu_o$ and $\nu_a$ are set to a higher value:

$$\nu_o = 1.76 \times 10^5 \text{ m}^2 \text{ s}^{-1}, \tag{27}$$
$$\nu_a = 1 \times 10^6 \text{ m}^2 \text{ s}^{-1}. \tag{28}$$



– dissipation-reducedstress

This experiment has the same parameters as the "dissipation" experiment, except for the coupling parameter $d$ which is reduced to $4 \times 10^{-8}$ s$^{-1}$.

Note that these idealised experiments do not take into account any dependence of the eddy (or turbulent) viscosity on the truncation scale as usually done in turbulence (e.g. Lesieur, 1990). However, even in the higher resolutions explored so far, we are still far from the scaling regimes for which these dependences may apply. In addition, the values of the eddy viscosity coefficients used are typically valid for a model configuration running at a spatial scale of the order of 100 km (Van der Avoird et al., 2002), which is smaller than the typical truncation used here. For this reason, we have performed a second experiment with a higher eddy viscosity coefficient. The problem of truncation and representation of subgrid-scale processes is an important open problem in climate modelling that needs careful attention. This matter falls beyond the scope of the present investigation, but forms the subject of a different study in context of the MAOOAM model (Demaeyer and Vannitsem, 2016). Note that in principle the dissipated kinetic energy should become an input to the thermodynamic equations of the system as positive heat source. As discussed in Lucarini and Fraedrich (2009), neglecting this process can have serious dynamical implications on long temporal scales. Additionally, an imperfect representation of this feedback between dynamics and thermodynamics is one of the sources of serious imperfections on the closure of the energy budget in climate models (Lucarini and Ragone, 2011; Lucarini et al., 2014). This issue will be analysed in future investigations.

## 4 Results

### 4.1 PUMA

Here we present the results for the two different experiments with PUMA, described in Section 3.3, and discuss our findings.

Figure 1 shows the two different Lyapunov spectra obtained in our experiments with PUMA. The averages were computed from a time series of 25 years of daily finite-time LEs. We can estimate the size of the attractor and by that estimate the degrees of freedom inherent to the attractor with the Kaplan-Yorke dimension $D_{KY}$, as described in Section 3.1. The number of positive exponents and $D_{KY}$ are shown in the legend of Fig. 1. Our findings confirm earlier results using two-layer QG models that suggested an increase of $D_{KY}$ and the number of positive Lyapunov exponents for a higher meridional temperature gradient (Lucarini et al., 2007; Schubert and Lucarini, 2015).

There are two very small exponents since the model setup is zonally symmetric which creates an additional zero mode (see Schubert and Lucarini (2015) for details). Otherwise, there are not many near-zero LEs in PUMA. There is continuity between the time scales that characterise the QG dynamics on the one hand, and faster, smaller-scale motions on the other hand. This shows that the usual assumption of a clear time-scale separation adopted when applying the filtering to derive the QG equations is, in fact, rather stretched with respect to reality.





Nevertheless, the 50 K spectrum in comparison to the 60 K spectrum has a much smaller slope where the LE are near zero and negative. This may suggest the presence a longer term regime switching behaviour. One potential source for such a regime change is the switching between blocked and non-blocked states of the mid-latitudes atmosphere.

We have computed the blocking rate employing the well established Tibaldi-Molteni Index (Molteni et al., 1988). We indeed
5  find a higher blocking rate for 50 K ($\approx 0.5\%$) than for 60 K ($\approx 0.25\%$). We would like to explore this connection further in future studies, especially in the direction of studying the location of the CLVs during blocking (Schubert and Lucarini, 2015).

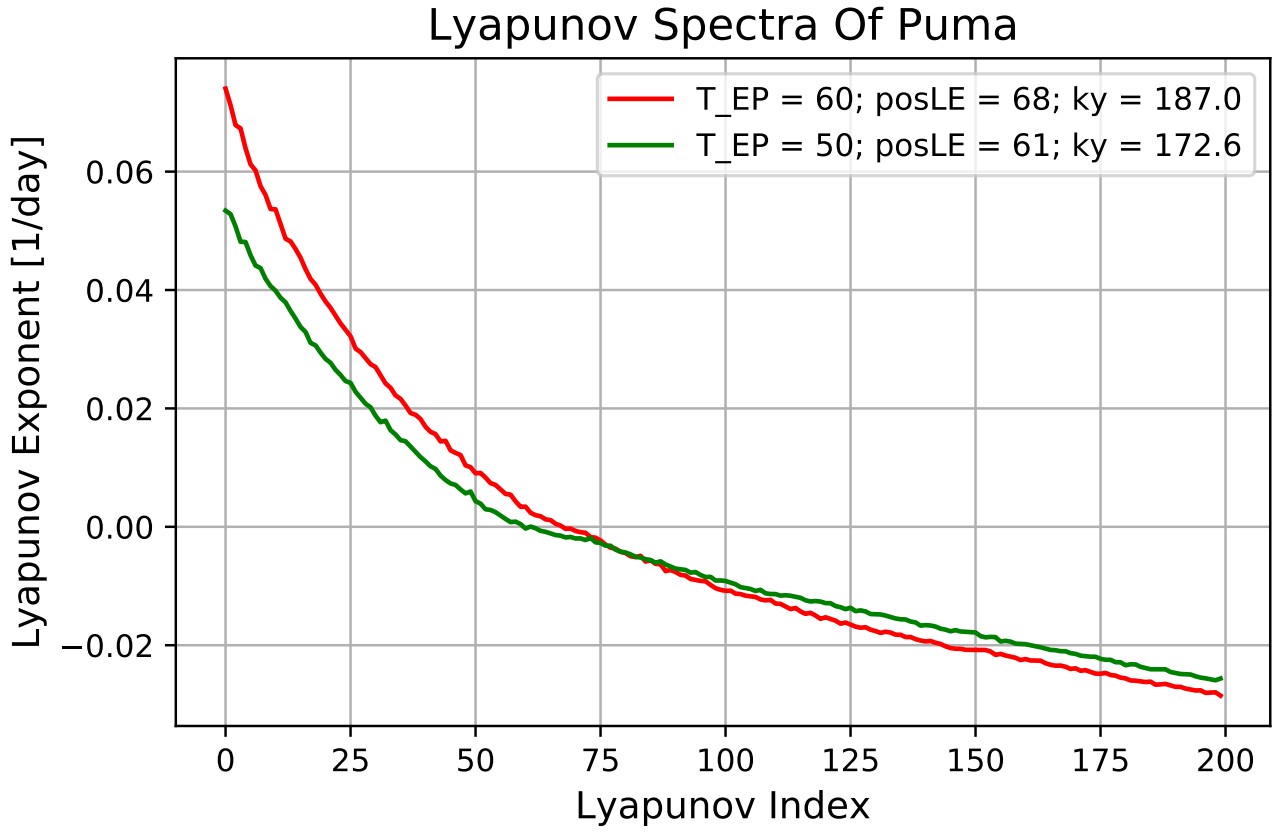

**Figure 1.** The Lyapunov spectra of PUMA for the two different setups with $\Delta T_{EP} = 50K$ and $60K$.

Next, the existence of a large deviation law for the FTLEs is verified, as described in Section 3.2. The decorrelation time $T_{\text{decorr}}$ is usually between 1 or 3 days. Therefore the rate function is computed for averages of length $tave = M \cdot 3$ days.

In Figs. 2 to 9 the results for the rate functions are shown for the fastest growing LE 1 (Figs. 2 and 3), faster decaying LE 150
10  (Figs. 4 and 5), the positive and negative near-zero LEs 59 and 64 for $\Delta T_{EP} = 50K$ (Figs. 6 and 7) and near-zero LEs 66 and 71 for $\Delta T_{EP} = 60K$ (Figs. 8 and 9). Since Eq. (22) is needed to compute the rate function, a long time series is necessary to estimate the distribution $P$ reliably. Our intent is to make at least a qualitative assessment of the convergence rate for $M \to \infty$.



The top panels of these figures show the approximation of the respective distributions obtained via kernel smoothing of the distribution of the block-averaged LEs. The bottom panels show the rate function for different $tave$ derived from Eq. (22).

We make the following observations. For all LEs the tendency for convergence of the rate function is visible. Also, the rate functions' shape is approximately parabolic and the estimates of the rate functions converge to the asymptotic with a comparable speed regardless of the value of the corresponding LE.

We interpret these results as another consequence of the non-existing clear-cut time-scale separation in a purely atmospheric model like PUMA. This is in opposition to what was speculated in Schubert and Lucarini (2015), where a primitive-equation model was expected to feature a time-scale separation visible in the Lyapunov spectrum. Such a time-scale separation would have been an a posteriori justification of the filtering by the QG approximation.

We have shown that in a primitive-equation model with a high-dimensional phase space of $\approx 60000$ the size of the attractor is small ($\approx 180$) in comparison. Nevertheless, the unstable subspace can still be regarded as a high-dimensional subspace. We also found sound results regarding the existence of a large deviation law independent of the growth rate of the linear perturbations. In hindsight with respect to the findings in MAOOAM this can be explained with the absence of a clear time-scale separation.

## 4.2 MAOOAM

The Lyapunov analysis is performed on the set of model configurations described in Section 3.4. Let us first evaluate the impact of the resolution on the amplitude of the dominant Lyapunov exponent. The largest Lyapunov exponent $\lambda_1$, which largely determines the limit of predictability, is plotted as a function of the model resolution for each experiment in Fig. 10. The dominant exponent $\lambda_1$ does not display a clear upward or downward trend versus model resolution and seems to stabilise for higher resolutions. This interesting feature suggests that the lower-order systems explored here already display a qualitatively correct amplitude for the dominant instability.

Furthermore, as we could expect, the predictability is enhanced for models where the scale-dependent dissipation term is present. The decrease in $\lambda_1$ also appears to suggest an enhanced predictability for models which have a larger ocean-atmosphere coupling parameter $d$, but this feature is not so clear for higher resolution versions. Vannitsem (2017) studied the dependence of the predictability on this coupling parameter in the low-order 36-variable model that lies at the basis of MAOOAM. Two distinct mechanisms were identified to drive the increase in predictability with increasing $d$. To a first approximation, the mechanical coupling of the fast atmosphere to the slow ocean corresponds to an effective friction term which reduces error growth in the atmosphere. Moreover, increasing the ocean-atmosphere coupling above a critical value induces a sudden jump in predictability, associated with the development of a slow coupled ocean-atmosphere mode (Vannitsem et al., 2015; Vannitsem, 2017).

Figures 11 to 13 show the full sets of Lyapunov exponents, or Lyapunov spectra, for the different experiments. These figures reveal the presence of three ranges in the spectrum of Lyapunov exponents: the positive, negative near-zero and large amplitude negative Lyapunov exponents, associated to the unstable, central and stable manifolds, respectively, in qualitative agreement with what was found in Vannitsem and Lucarini (2016). We expect that the stable and unstable manifolds mainly characterise



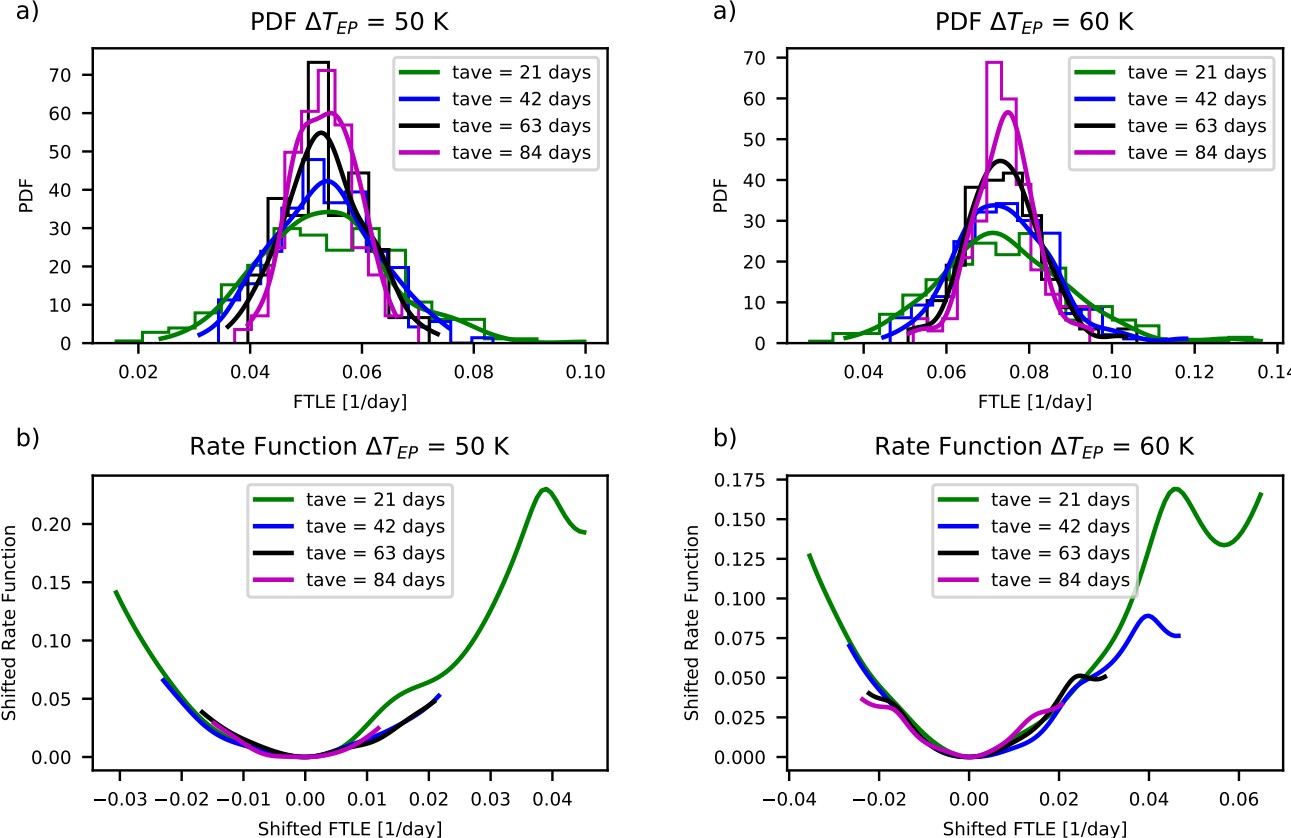

**Figure 2.** Distributions and rate functions of $\lambda_1$, the fastest-growing instability in PUMA, for $\Delta T_{EP} = 50K$. The top panel shows the different distributions and their kernel-smoothing approximation of $\sigma_M^1$ where $tave$ is the respective $3\,M$. The bottom panel shows a comparison of the rate functions, with the minimum moved to zero.

**Figure 3.** Distributions and rate functions of $\lambda_1$, the fastest-growing instability in PUMA, for $\Delta T_{EP} = 60K$. Panels as in Fig. 2.





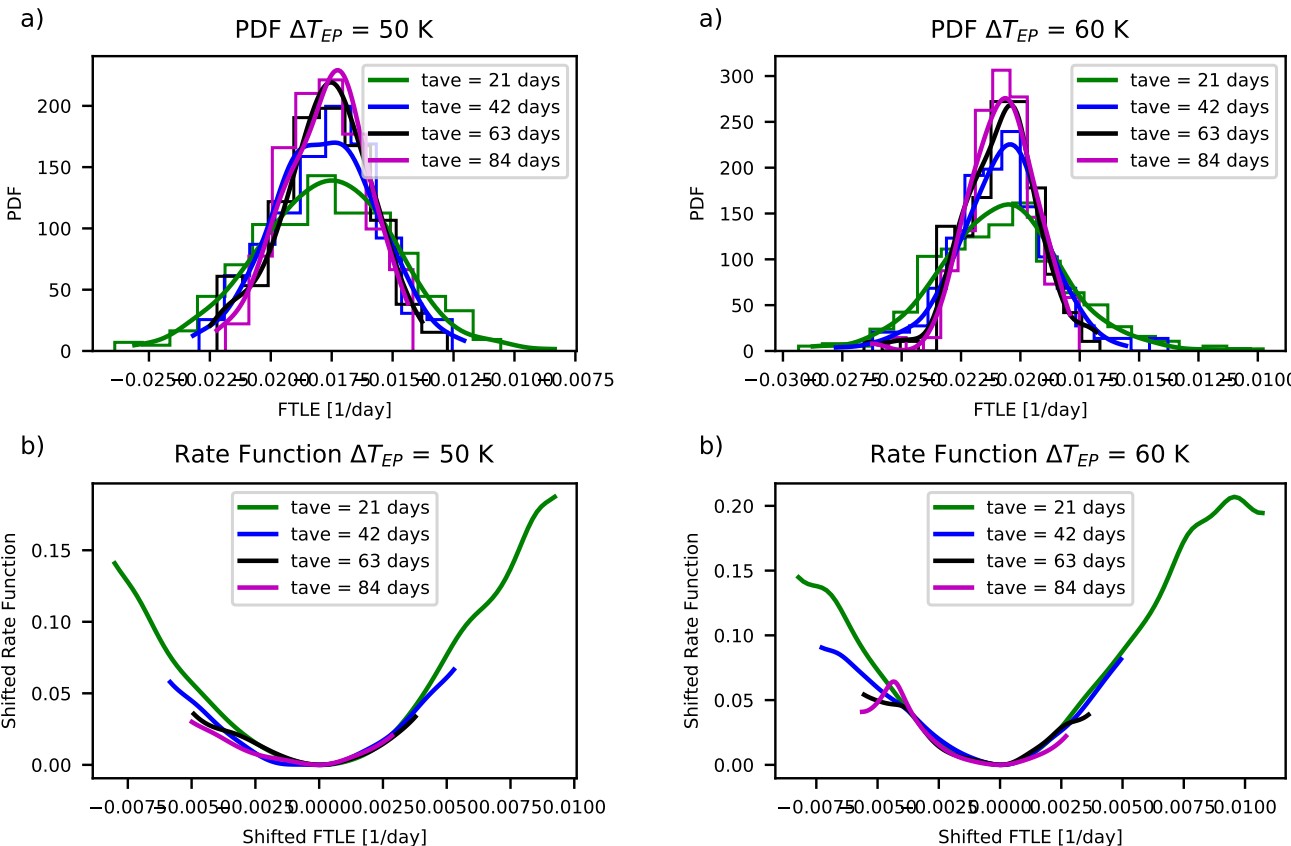

**Figure 4.** Distributions and rate functions of $\lambda_{150}$, a strongly decaying direction in PUMA, for $\Delta T_{EP} = 50K$. Panels as in Fig. 2.

**Figure 5.** Distributions and rate functions of $\lambda_{150}$, a strongly decaying direction in PUMA, for $\Delta T_{EP} = 60K$. Panels as in Fig. 2.

the dissipative and unstable motions of the atmosphere, while the central manifold also projects considerably on the variables of the ocean.

The highly populated central manifold of MAOOAM is in stark contrast with the few near-zero LEs in PUMA. Being a purely atmospheric model, PUMA's Lyapunov spectrum does not exhibit the large time-scale separation present in MAOOAM. Indeed, the spectrum of PUMA bears more resemblance to that of the QG two-layer model of Schubert (2015).

Upon increasing the number of modes in the ocean and the atmosphere, the number of positive Lyapunov exponents (indicated with a vertical arrow) consistently increases, but not as much as the number of strongly negative exponents. This suggests that most of the additional spatial scales that are resolved by the higher-resolution models are highly dissipative, hence increasing the number of strongly negative Lyapunov exponents. The additional positive and near-zero exponents that are introduced at these scales nevertheless indicate that the added resolution still resolves some scales that are important for the description of the dynamics. This is in agreement with the conclusion in De Cruz et al. (2016), where it was shown that in order to describe



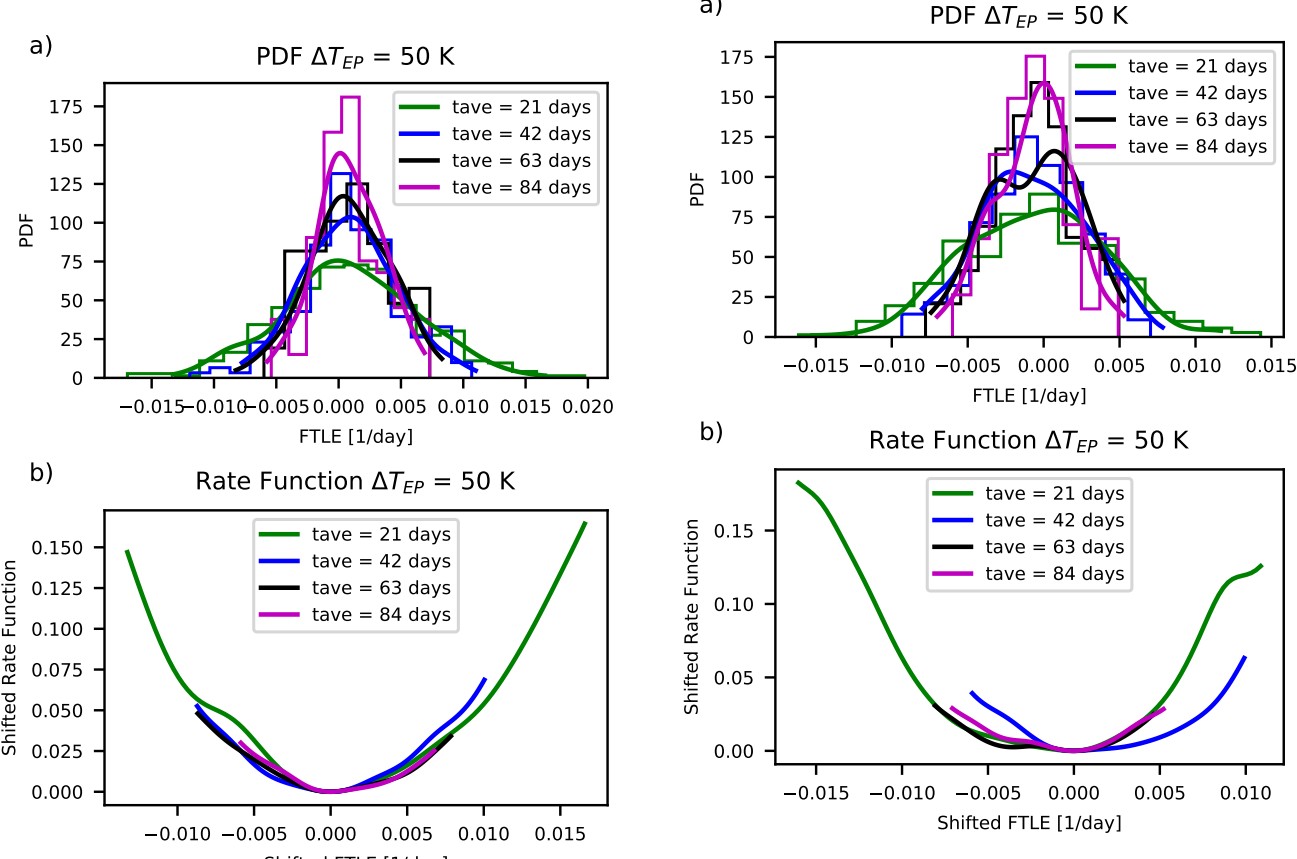

**Figure 6.** Distributions and rate functions of $\lambda_{59}$, a **near-zero, growing** instability in PUMA, for $\Delta T_{EP} = 50K$. Panels as in Fig. 2.

**Figure 7.** Distributions and rate functions of $\lambda_{64}$, a **near-zero, decaying** instability in PUMA, for $\Delta T_{EP} = 50K$. Panels as in Fig. 2.

the ocean dynamics, one needs to be able to resolve the Rhines scale $L_{Rh} = \sqrt{\frac{U}{\beta}}$, requiring oceanic wavenumbers as high as of 40–50.

Figure 15 plots the Kaplan-Yorke dimension $D_{KY}$ as a function of the model resolution. This shows that $D_{KY}$ is the highest for the models which do not include the scale-dependent dissipation process (nodissip). A reduction in the ocean-atmosphere coupling $d$ appears to slightly increase $D_{KY}$ for most model resolutions, both in the case with and without scale-dependent dissipation. The tenfold increase in the dissipation parameters $\nu_o$ and $\nu_a$ (dissipationx10) results in the lowest values for $D_{KY}$, as can be expected from a more dissipative but still chaotic system.

As the number of dimensions increases quadratically and not linearly for the consecutive model resolutions, it is instructive to rescale $D_{KY}$ by the number of dimensions $N$, as shown in Fig. 16. This shows that while $D_{KY}$ increases with resolution, the attractor dimension's fraction of the full phase space dimension decreases (even if slowly) with increasing resolution from the



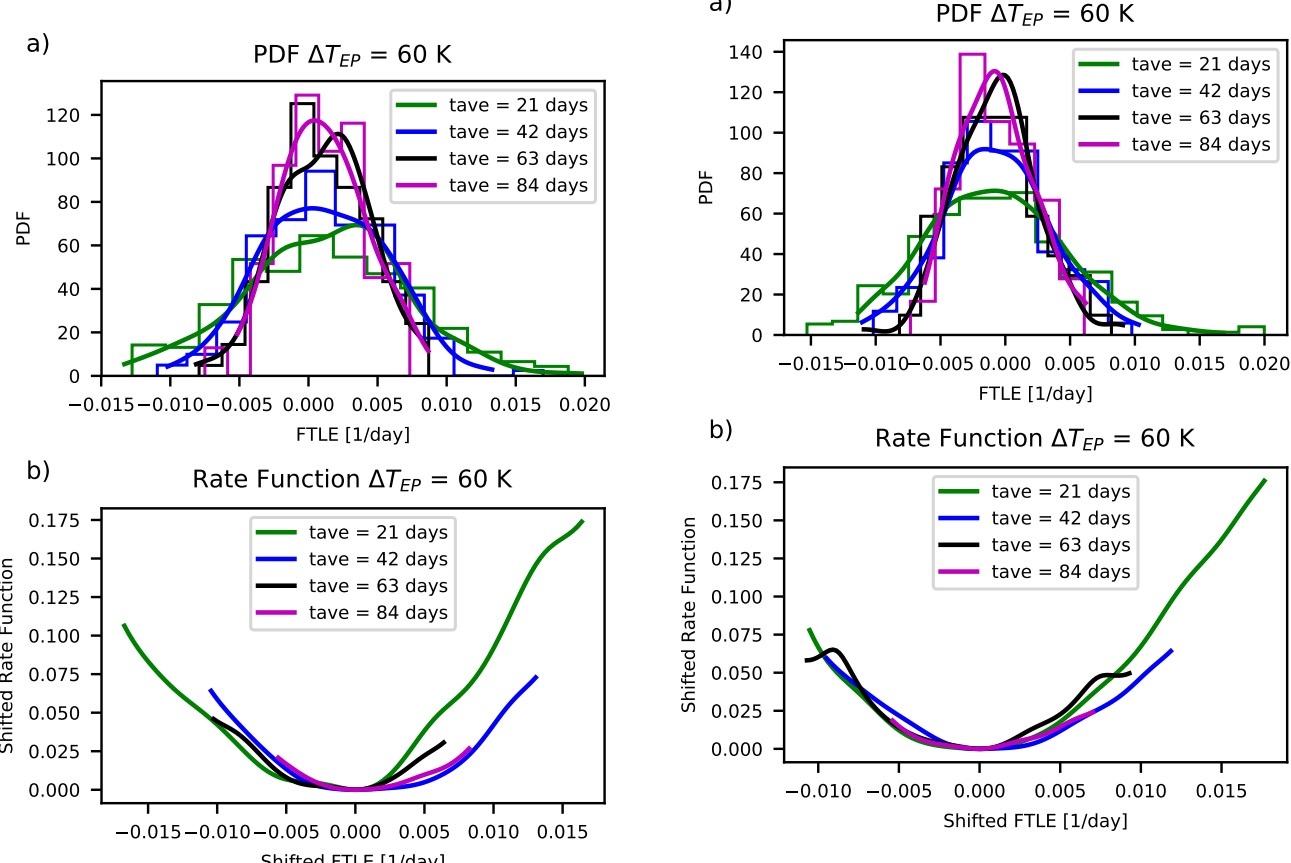

**Figure 8.** Distributions and rate functions of $\lambda_{66}$, a **near-zero, growing** instability in PUMA, for $\Delta T_{EP} = 60K$. Panels as in Fig. 2.

**Figure 9.** Distributions and rate functions of $\lambda_{71}$ a **near-zero, decaying** instability in PUMA, for $\Delta T_{EP} = 60K$. Panels as in Fig. 2.

atm. 6x6, oc. 6x6 models onward, for all experiments, suggesting that one is adding in higher proportion highly stable modes that do not necessarily play an important role in the dynamics. In other words, we are not in the regime where the system is extensive, as, in fact, the geometry of the domain is fixed and we are capturing a larger and larger (yet insufficient) fraction of the active dynamical processes as the resolution is increased. Had we reached the optimal resolution, Fig. 15 would be flat, and

5  Fig. 16 would approach zero.

Figure 17 shows the Kolmogorov-Sinai entropy $h_{KS}$ versus model resolution, for the different experiments. The trends for the "nodissip" and "nodissip-reducedstress" experiments appear to suggest that $h_{KS}$ would increase unboundedly for increasing model resolution if a parametrization for the scale-dependent dissipation is absent. The experiments which take this process into account, paint a more realistic picture, with $h_{KS}$ levelling off at the highest model resolutions.


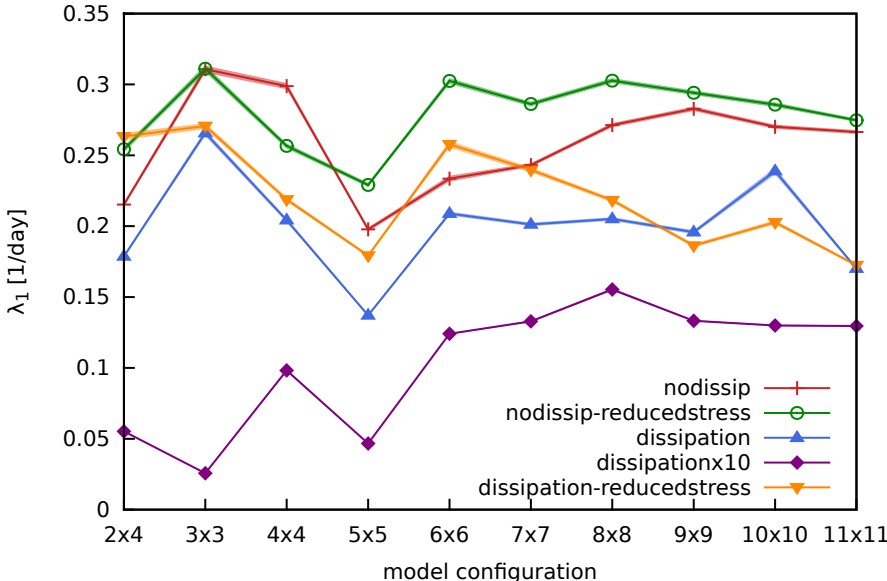

**Figure 10.** The largest Lyapunov exponent $\lambda_1$ of MAOOAM as a function of the model resolution for the different experiments: dissipation (red pluses), dissipation-reducedstress (green circles), dissipationx10 (blue upward-pointing triangles), nodissip (purple diamonds), nodissip-reducedstress (orange downward-pointing triangles).

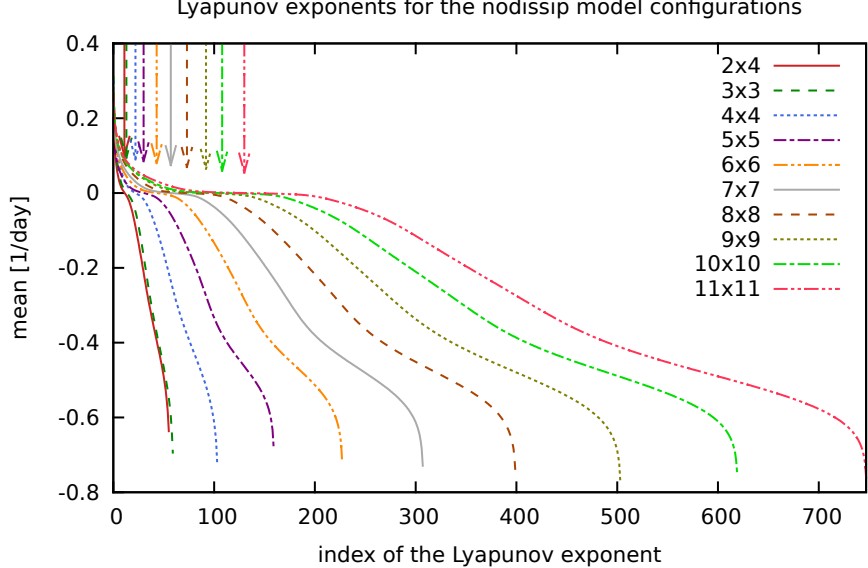

**Figure 11.** Lyapunov spectra of MAOOAM for the "nodissip" experiment, for model configurations from atm. 2x4, oc. 2x4 (red full line) up to atm. 11x11, oc. 11x11 (pink dash-dot-dotted line). Lyapunov exponents are ranked in decreasing order, and the index of the smallest positive Lyapunov exponent is indicated with a downward-pointing arrow for each model configuration.


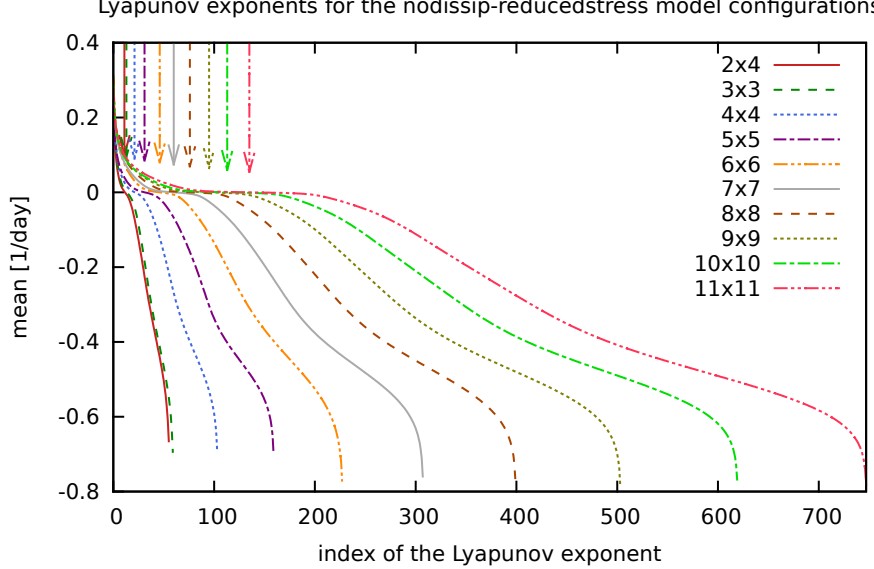

**Figure 12.** Lyapunov spectra of MAOOAM for the "nodissip-reducedstress" experiment. Colours and arrows as in Fig. 11.

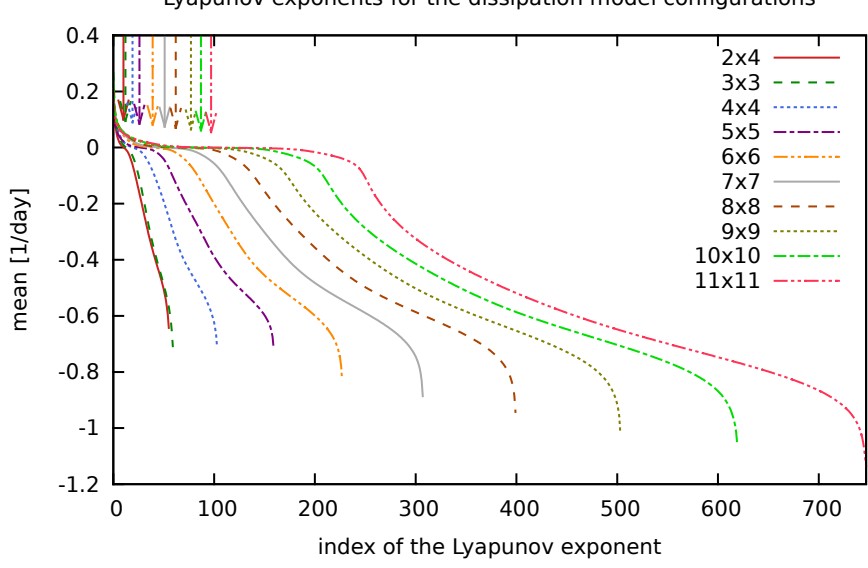

**Figure 13.** Lyapunov spectra of MAOOAM for the "dissipation" experiment. Colours and arrows as in Fig. 11.

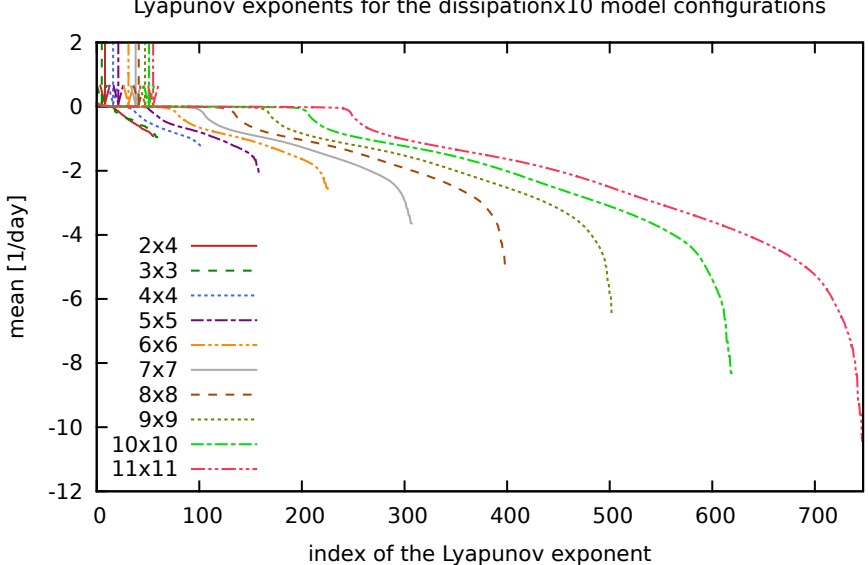

**Figure 14.** Lyapunov spectra of MAOOAM for the "dissipationx10" experiment. Colours and arrows as in Fig. 11.

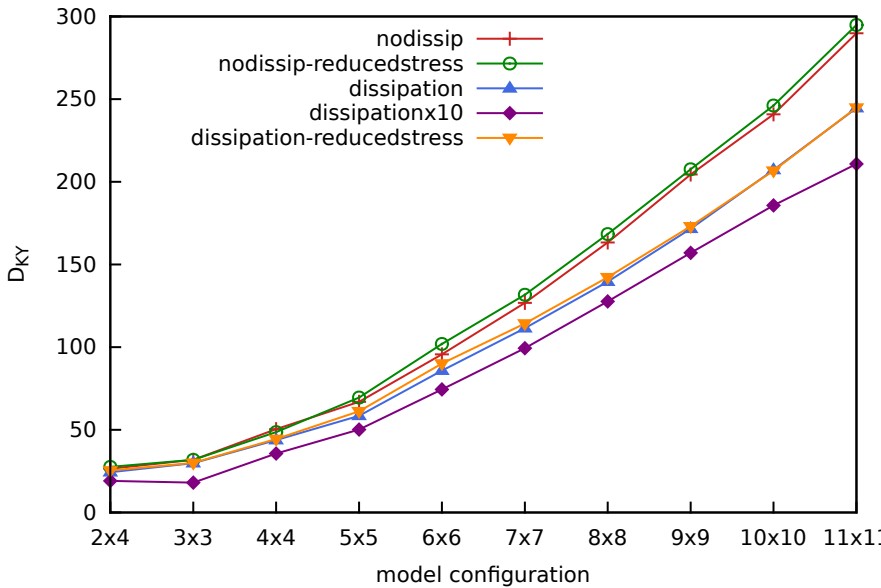

**Figure 15.** Kaplan-Yorke or Lyapunov dimension $D_{KY}$ of MAOOAM as a function of the resolution for the different experiments. Colours as in Fig. 10.

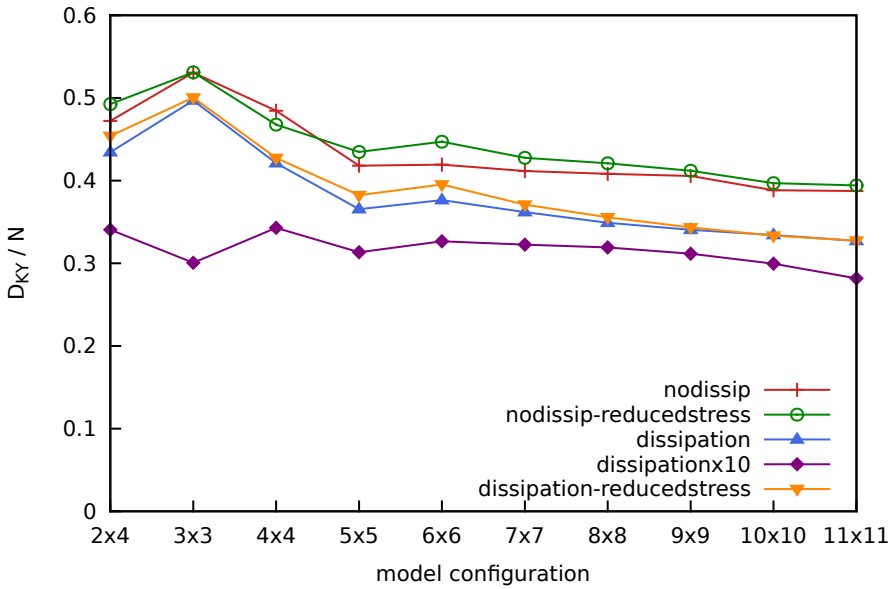

**Figure 16.** Kaplan-Yorke or Lyapunov dimension $D_{KY}$ of MAOOAM divided by the total number of dimensions $N$, as a function of the resolution for the different experiments. Colours as in Fig. 10.

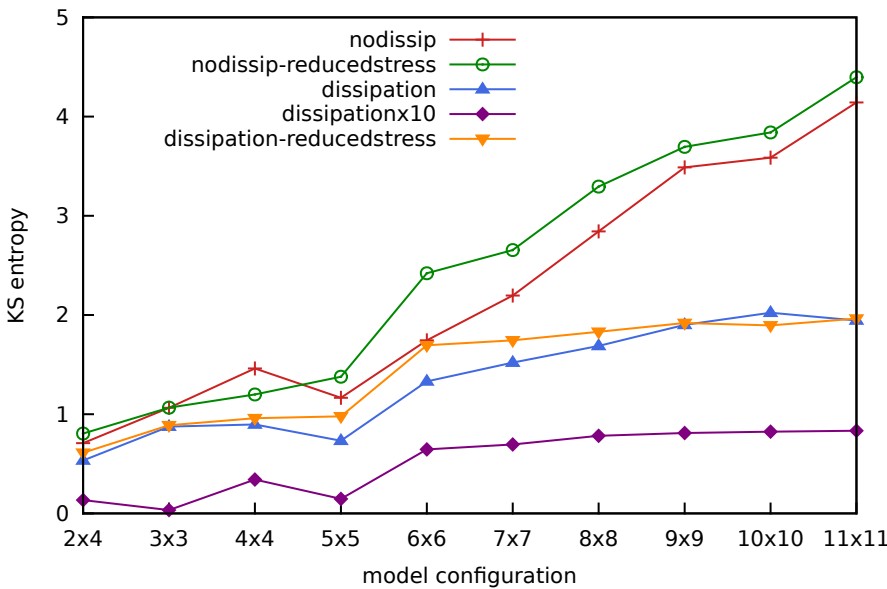

**Figure 17.** Kolmogorov-Sinai entropy $h_{KS}$ of MAOOAM as a function of the resolution for the different experiments. Colours as in Fig. 10.





An additional experiment is performed by increasing the resolution of the ocean and of the atmosphere separately starting from a specific symmetric configuration "6x6". Figure 18 displays the Lyapunov spectra for the model configuration "dissipation". Two important features stand out: (i) when the resolution of the atmosphere is increased, the majority of the new exponents populate the stable manifold; (ii) on the contrary when the resolution of the ocean is increased, the number of slightly positive and slightly negative exponents increases considerably. This also suggests that the increase of Lyapunov dimension and the number of positive exponents after a resolution of atm. 6x6 - oc. 6x6 should be attributed to the presence of the ocean. In this sense the ocean plays an active role in the development of the coupled dynamics. This result deserves an extensive investigation by looking at the properties of the CLVs.

As a final analysis, we have given a preliminary look at whether large deviation laws can be established for the long-term statistics of the FTLEs. In what follows, we consider the "9x9" simulations. Similarly to what was found in a previous analysis performed on a severely truncated version of MAOOAM (Vannitsem and Lucarini, 2016), we find that the time series of the FTLEs corresponding to the strongly damped mode are weakly correlated and one can construct the rate functions defining the large deviations laws; compare figures 19a-c) for the $351^{st}$ LE. Additionally, the lagged time correlation of the near-zero LEs are very strong and it makes no sense to look for large deviation laws in this case.

In contrast to what was presented in Vannitsem and Lucarini (2016), establishing large deviation laws for the FTLEs associated with positive LEs is not trivial, even when one considers the first FTLE. Lagged-time correlations are such that the available time series are not sufficiently long to reach the asymptotic limit, except for the *nodissip* simulation scenario, which allows for the presence of a larger value of the $1^{st}$ LE and faster decay of correlations; compare Figs. 20a)-c). This suggests that when many unstable modes are present, disentangling their long-term properties requires very long integrations, possibly as a result of geometrical quasi-degeneracies among such modes. This is an issue that should be further explored, given its practical and theoretical relevance. One can conjecture that the damped modes do not feature such properties as their dynamics is mostly driven by linear dissipative processes. Therefore, we propose that an accurate analysis of the tangent space with the formalism of CLVs is required to advance our understanding of predictability at medium and long time scales.

In brief, these results indicate that the dominant instabilities of the coupled ocean-atmosphere system are well captured by MAOOAM, even at low resolutions. However, the increase of the Lyapunov dimension with the resolution implies that the relevant dynamics of the system are not yet fully resolved, in agreement with De Cruz et al. (2016). The main role of the ocean in this matter is confirmed by varying the ocean and atmosphere resolutions independently. Conversely, increasing the resolution of the atmosphere only adds highly dissipative modes. Finally, in contrast with what was found for a low-order version of MAOOAM (Vannitsem and Lucarini, 2016), large deviation laws cannot be established for the near-zero and positive FTLEs in the "9x9" configuration.

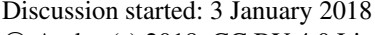



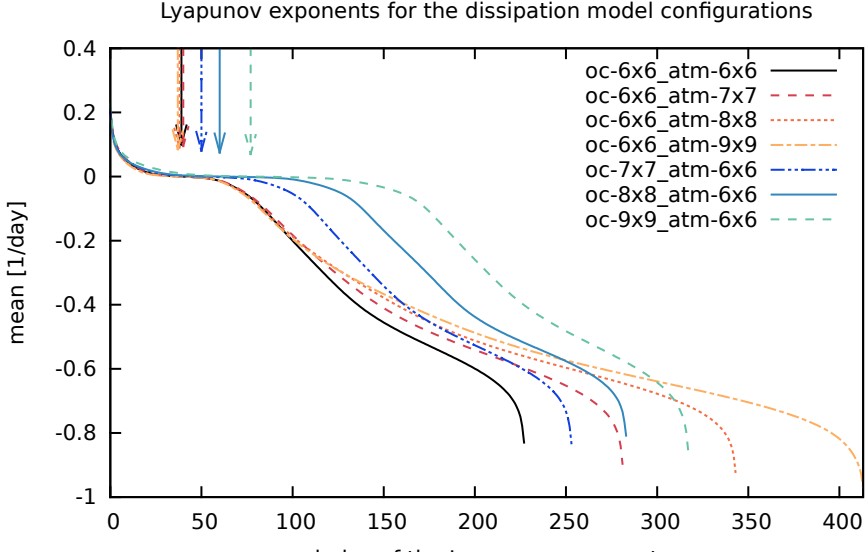

**Figure 18.** Lyapunov spectra of MAOOAM for the "dissipation" experiment, for different model configurations starting from atm. 6x6, oc. 6x6 (black full line). The resolutions of the ocean (dark to light blue lines) and the atmosphere (red to orange lines) are modified separately. Lyapunov exponents are ranked in decreasing order, and the index of the smallest positive Lyapunov exponent is indicated with a downward-pointing arrow for each model configuration.

## 5   Toward a new programme

The chaotic nature of the atmosphere and of the climate system has been investigated in the present work in the context of a primitive-equation atmospheric model and a coupled ocean-atmosphere model. Both systems suggest that high-dimensional dynamical processes are at play with very interesting distinct specificities.

The Lyapunov spectra of the two models considered here have rather different qualitative features, as a result of their structural differences, which have profound impacts on the type of possible instability mechanisms. Following Gallavotti and Lucarini (2014), one expects that if a clear time-scale separation between distinct dynamical regimes is present, one should find that the Lyapunov exponents can be divided into separate groups, corresponding to distinct clusters in their values. This is the analogue in full nonlinear terms of what is envisioned by the usual scale analysis of GFD equations.

MAOOAM is a coupled quasi-geostrophic atmosphere-ocean model, which, by definition, features a large time-scale separation between ocean and atmosphere, and lacks a satisfactory representation of mesoscale and sub-mesoscale processes. PUMA is an atmospheric-only primitive-equation model, which can represent the faster, smaller-scale instabilities associated with processes occurring well below the Rossby deformation radius. On the other side, the lack of an active ocean component removes the presence of very slow scales and does not allow for a built-in scale separation in the dynamics.

We summarise here some findings:



- In PUMA the spectrum of Lyapunov exponents changes in accordance with the paradigm that stronger baroclinic forcing leads to a more unstable atmosphere, as already observed in Schubert and Lucarini (2015) for a quasi-geostrophic model. The model does not feature any separation of scale, as the Lyapunov spectrum is quite smooth. As a result, one cannot clearly distinguish the modes corresponding to baroclinic instability, Kelvin-Helmholtz instability, etc. Despite this, we find that for the lower meridional temperature gradient ($\Delta T_{EP} = 50K$) the spectrum features more negative closer to zero exponents. Interestingly, this might be related to the presence of blocking, but specific studies with more robust dynamics between blocking and non-blocking situations are necessary to clarify that. Additionally, one finds that all the FTLEs accurately obey large deviation laws defining the predictability properties at long time scales, including the near-zero exponents. The model can be categorised as being nonuniformly hyperbolic with a trivial central manifold (the direction of the flow).

- For MAOOAM the Lyapunov spectrum is shaped considerably by the presence of the ocean, with a large portion of exponents close to zero. The subspace associated with these exponents corresponds to the central manifold as in the theory of partially hyperbolic systems, and presents features analogous to what was observed in Vannitsem and Lucarini (2016). Furthermore, raising the ocean resolution in MAOOAM clearly increases the number of both positive and negative near-zero Lyapunov exponents, which implies a considerable increase of the Lyapunov dimension of the attractor. Reducing the intensity of the dissipative processes leads, as expected, to an increase in the instability of the model.

  One can also conjecture that the set of physical modes, as defined by Yang and Radons (2013), are not yet fully populated since one would expect that the isolated modes are strongly dissipative. This might imply that the resolution necessary to correctly describe the dynamics of the system is much higher in the ocean. This aspect is well known in ocean dynamics since the unstable baroclinic modes, that play an important role in the ocean variability, can only be resolved with scales smaller than 50 km. Yet the question of defining an appropriate resolution (or more exactly an appropriate set of dynamical modes) for which the dynamics is well captured is still open and the analysis of the CLVs in the spirit of (Yang and Radons, 2013; Vannitsem and Lucarini, 2016) can help answer this very important question.

  The analysis of the FTLEs of MAOOAM reveals some interesting insight into the dynamics. The FTLEs associated to the strongly dissipative modes obey large deviation laws, while those corresponding to the near-zero LEs do not. This behaviour is expected, and in agreement with what was found in Vannitsem and Lucarini (2016). Surprisingly, however, it is hard to find convergence for the FTLEs associated to the positive LEs. This may point to the presence of nontrivial ocean influence on the (mostly) atmospheric instabilities.

In the programme we want to develop starting from this investigation, we will employ CLVs in high-dimensional models to tackle various open problems. CLVs allow to associate growth and decay rates to time-dependent physical modes, and provide a geographical portrait of where instability or damping develops.

First, what is the minimal but sufficient resolution? This is a crucial question, in particular in view of the current computer power needed to perform long-term numerical integrations. A possible way to quantify where this threshold might be, is by means of the different modes identified by Yang and Radons (2013) using CLVs. The CLVs provide information on the optimal



splitting of *physical modes* that effectively describe the dynamics of the system and the highly damped modes. The latter can be considered as noisy, purely dissipative terms whose resolution is not necessarily relevant, and are also called *isolated modes*. Yang and Radons (2013) interpreted them as the result of having a larger number of degrees of freedom in the model than required to resolve all meaningful physical processes. The central feature that allows for the splitting is the angle between

the CLVs. If two CLVs display angles around 90 degrees and bounded away from zero degrees, these directions in phase space can be naturally split. Being able to describe the physical modes is deemed essential to satisfactorily reproduce the so-called inertial manifold. The inertial manifold contains the effective finite-dimensional dynamics of the system, which, we remind, is originally infinite-dimensional if we represent a continuum system described by (S)PDEs. In particular, it would be interesting to determine how the threshold for resolving the inertial manifold varies in a purely atmospheric model compared

to a coupled model atmosphere-ocean model. Additionally, the analysis of geometry of the tangent space can clarify to what extent a system can be treated - effectively, not rigorously - as hyperbolic vs partially hyperbolic. As explained in Vannitsem and Lucarini (2016), this has profound implications for the predictability.

Second, we want to understand multiscale instabilities better and find out what are the driving processes behind their growth. Here, the covariance of the CLVs with the tangent linear equation is the key for understanding instabilities and their properties

far away from an equilibrium. Traditionally, even in a chaotic setting such an analysis relied on classic normal mode instability of fixed stationary states (e.g. Charney, 1947; Eady, 1949; Pedlosky, 1964) to explain phenomena like the baroclinic and barotropic instability. This approach has been very beneficial but has many known shortcomings for the understanding of highly nonlinear phenomena such as wave-wave interactions (Speranza and Malguzzi, 1988) or regime switching like blocking (Pelly and Hoskins, 2003). Additionally, CLVs will allow us to better understand coupled ocean-atmospheric modes. We wish

to develop our future programme in line with Schubert and Lucarini (2015, 2016) who demonstrated that CLVs give a picture of what types of instabilities exist in an atmospheric quasi-geostrophic (QG) two-layer model and of the energetics behind them. For example, the fastest modes can be almost exclusively barotropically unstable even though traditional normal mode analysis suggests the most unstable modes are driven by baroclinic energy conversion. Given these findings, we expect an even more diverse mixture of different types of instabilities in multiscale systems such as PUMA or MAOOAM. This approach is a

promising alternative to restricting the analysis to either studying idealised life cycles of instabilities (Plougonven and Zhang, 2014) or studying yet again normal modes (Molemaker et al., 2005).

*Code availability.* The PUMA model is a part of PLASIM, for which the source code can be downloaded at https://www.mi.uni-hamburg.de/en/arbeitsgruppen/theoretische-meteorologie/modelle/sources/plasim.tgz. The source code for MAOOAM v1.2 is available at http://github.com/Climdyn/MAOOAM. This version is also archived at http://dx.doi.org/10.5281/zenodo.231162. The source code to compute the Lya-

punov exponents is available upon request to the corresponding author.





*Data availability.* The Lyapunov spectra of the different PUMA and MAOOAM model configurations, that were computed using the Benettin algorithm, are available as supplementary material.

*Author contributions.* V. Lucarini and S. Schubert performed the analysis of PUMA. S. Schubert wrote the code to compute the LEs for PUMA. L. De Cruz and S. Vannitsem performed the analysis of MAOOAM. J. Demaeyer and S. Schubert wrote the code to compute the LEs
5  for MAOOAM. L. De Cruz, S. Vannitsem and J. Demaeyer wrote MAOOAM. V. Lucarini analysed the FTLEs in MAOOAM. All authors contributed to the writing of the manuscript.

*Competing interests.* S. Vannitsem and V. Lucarini are members of the editorial board of the journal. The other authors declare that they have no conflict of interest.

*Acknowledgements.* LDC was supported by the BELSPO under contract BR/165/A2/Mass2Ant. JD was supported by BELSPO under con-
10  tract BR/121/A2/STOCHCLIM. VL and SS were supported by the DFG through the contract SFB/Transregio TRR181.



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

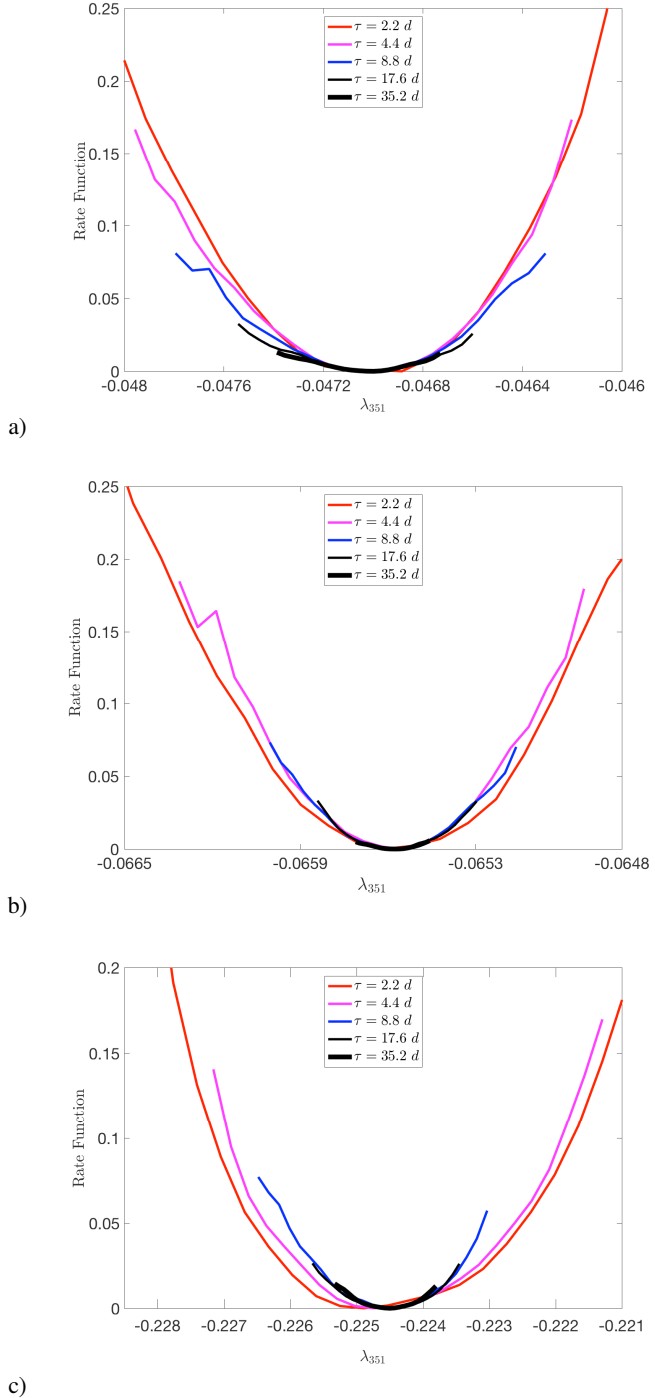

a)

b)

c)

**Figure 19.** Estimate of the rate function describing the large deviation law of the $351^{st}$ FTLE for the MAOOAM model with no dissipation (a), reference value for the dissipation (b), and enhanced dissipation by a factor of 10 (c). Convergence is apparent only in a).





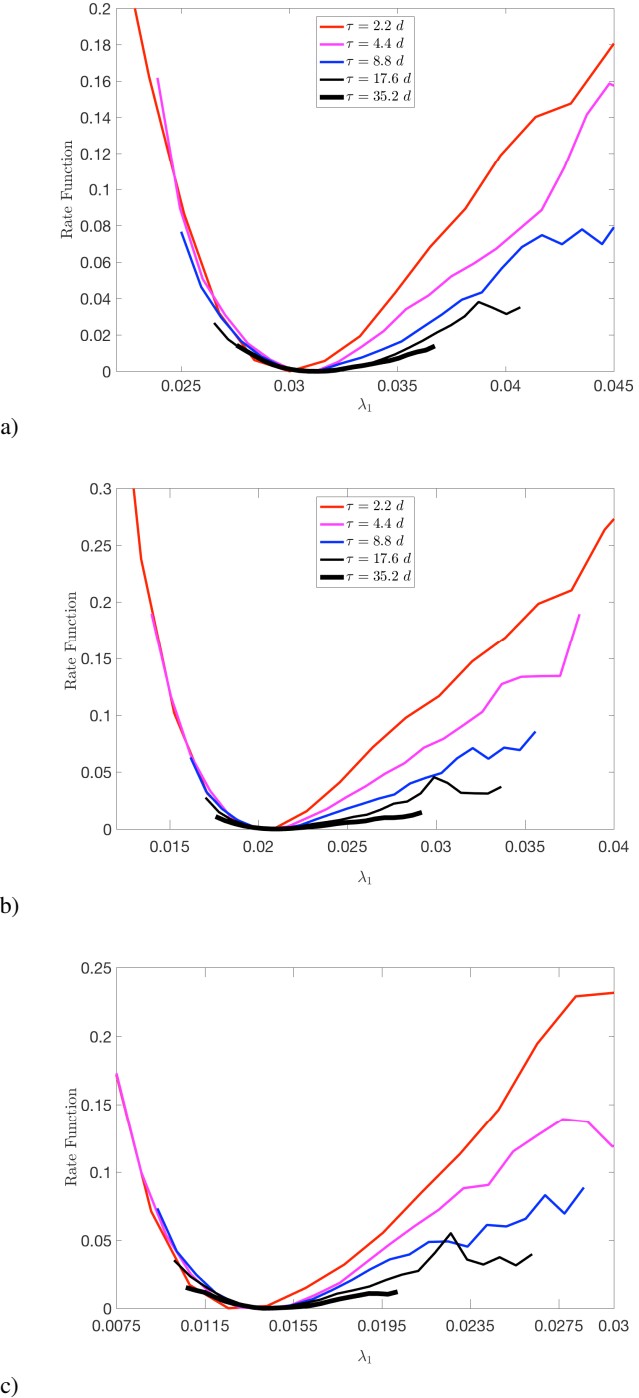

**Figure 20.** Estimate of the rate function describing the large deviation law of the first FTLE for the MAOOAM model with no dissipation (a), reference value for the dissipation (b), and enhanced dissipation by a factor of 10 (c). Convergence is apparent only in a).