# Peer review of "Exploring the Lyapunov instability properties of high-dimensional atmospheric and climate models"

_Nonlinear Processes in Geophysics, 2017_

## Referee Comment (RC1) · Anonymous Referee #1 · 5 Feb 2018

This manuscript presents a Lyapunov analysis of two models: PUMA (purely atmospheric) and MAOOAM (atmosphere-ocean coupled). The purpose of this is to investigate the impact of different configurations (resolution, dissipation, and atmosphere-ocean coupling) on the instabilities of the system. This is an interesting path of research since, ultimately, it will permit to understand better the goodness of a certain model for forecasting certain processes operating at particular spatio-temporal scales.

While I consider the topic of this work is worth of being published, and the manuscript reasonably well written, I have one important criticism on the methodological procedure that prevents me from recommending its publication.

An important part of the manuscript is devoted to the rate function of the finite-time Lyapunov exponent (FTLE) distribution. As claimed in the abstract

1) For the PUMA model: "The convergence rate of the rate function(al) for the large deviation law of the FTLEs is fast for all exponents".

2) For the MAOOAM: "[...] it is possible to robustly define large deviation laws describing the statistics of the FTLEs corresponding to the strongly damped modes, [...]"

My main criticism is the meaningfulness of the rate function analysis considering the data used and their quality. I itemize next my concerns (a-c):

a) In all cases the claimed convergence is far from apparent with the naked eye. In my view the rate functions vary consistently as parameter tave changes, but I don't see a true convergence of the data as such. I find very questionable sentences like "For all LEs the tendency for convergence of the rate function is visible" in page 16, line 3; and "...FTLES accurately obey large deviation laws ..." in page 27. To judge the convergence from the shift of a whole curve as a parameter changes is really problematic.

b) The data analysis resorted to a strong smoothing of the data obtained from short time series, cf. the histograms. It is difficult to evaluate the errors accumulated thereby.

c) From a theoretical point of view, I have doubts that the rate function can be detected with the time intervals over which the FTLEs are computed ("tave"). I'm afraid that the values of "tave" used are simply too small to reach the asymptotic rate function, even if the time series were infinitely long. From a theoretical perspective the choice of a reference time "T", based on the time interval in which autocorrelations of the FTLE decay below 1/e, presents certain problems. The rate function is calculated for times only up to 28*T, what may be too small to detect the rate function (independently of the amount of data). I say this because in many spatio-temporal chaotic systems the "renewal time" of the Lyapunov vector operates at a time scale $T_x$ much larger than T.

In such a case, in order to detect the rate function one must take time intervals (tave) larger than $T_x$ ($>> T$), see Pazo et al (2013). For instance, in Laffarge et al (2013), with 40 coupled maps the time interval (tave) used to measure the rate function is $10^4$ iterations.

The only way I see to demonstrate something unambiguously with the numerical data at hand is to check the convergence of a particular quantity (instead of a curve). Following Pazo et al (2013), and as double-check, I suggest to measure the variance of the FTLE for different "tave" values. Multiplying by tave, it should be possible to verify if the data level off at a certain value D in the range of tave values considered. (The diffusion coefficient D is the inverse of the second derivative of the rate function at its minimum, see e.g. (Kuptsov and Politi, 2011)).

Minor comments:

1. Eq. (19), please mention that $M^*$ is the adjoint of M. Note also that M has a wrong font type in Eq. (19).

2. Letter M is used for the resolvent matrix and for the integer (tave= T*M). I would avoid this duplicity.

3. The concept backward Lyapunov exponent appears in page 10, line 22, without much explanation. Note that Lyapunov exponents obtained from Eq. (19) are actually "forward Lyapunov exponents". The mirror definition of Osedelets theorem with $M M^*$, instead of $M^* M$, yields backward LEs. I point to table 1 in Pazo et al (2010) and to (Ershov & Potapov, 1998. On the concept of stationary Lyapunov basis. Physica D 118(3-4), 167–198.) for the formal link between Osedelets theorem and Bennetin's algorithm.

4. Page 11. The relationship of the KY dimension with the fractal dimension was confusing for me as written now (probably due to the intention of making it simple for the unfamiliar reader). The KY dimension is an estimation of the information dimension,

usually denoted D_1. D_1 is known to be (equal or) smaller than the capacity or box-counting dimension (D_0), which I guess is what the authors refer to by "fractal dimension", following the paper by Frederickson et al (1983). My taste is that nowadays one can talk of D_KY as estimation of D_1, and perhaps to mention the information dimension bounds the capacity/fractal dimension (for the unfamiliar reader on these questions).

5. Page 11, when introducing the FTLEs the authors refer to Haller (2000). I have nothing against, but I think it is more appropriate H. Fujisaka, Prog. Theor. Phys. 70, 1264 (1983)

6. Page 15, line 1, I think the use of "much smaller" is exaggerating the difference between the spectra. Using "smaller" is enough.

7. Page 15, line 4, when mentioning the Tibaldi-Molteni index, why is not the paper of Tibaldi and Molteni (1990) cited instead?

8. It should be said somewhere that Fig. 1 shows (only) the 200 largest LEs.

9. In Fig. 1, it looks like the Lyapunov index starts at 0, instead of 1. I guess the lines have to be displaced 1 unit rightwards.

10. Page 15, line 9, "faster" -> "fast" (?)

11. In Fig. 2 it is not said which kernel function is used, or how the bandwidth was optimized.

12. Figs. 4,5 the x-tic labels overlap.

13. Figs. 11-14. I'm curious how the Lyapunov spectra look like when the x-coordinate (the Lyapunov index) is rescaled by the number of degrees of freedom. Is there some overlapping of the data for the most negative LEs?
* * *
2017-76, 2018.

---

## Referee Comment (RC2) · S.G. Penny (Referee) · 10 Feb 2018

General points:

In general, this is a very nicely written paper. The introduction is accessible and informative. The results are interesting and I believe will inspire a number of new research directions.

I'd suggest that the authors perform a forced atmosphere-only and a forced ocean-only experiment with the MAOOAM system to compare how typical forced atmosphere or ocean models compare to coupled systems in terms of the Lyapunov spectrum. Or if

it already exists, point to a previous work by the authors that has done this comparison. This would be helpful for guidance to the operational centers currently making decisions about what is gained from transitioning from separate component forecast systems to a fully coupled forecast system.

As a general comment, the authors should strive to cite the original works for various concepts rather than a reference text or review paper.

To the editor: Regarding the journal's typesetting decisions, please place the figures closer to where they are mentioned in the text.

Technical points:

L 9-10:

"but also the errors that are present either in the model parametrizations, known as model errors,"

It would be more accurate to say this is known as 'model parameterization error', since 'model error' includes systematic misrepresentation of the system dynamics.

L 11:

Missing period at the end of the line.

L 30:

This paragraph should start with a "However,"

L 32:

" for atmospheric instabilities, and most notably convective"

Remove the word 'and'

L 34:

"The oceanic circulation, by contrast, is mostly mechanically driven by atmospheric winds"

This is true on shorter timescales, but you should also acknowledge buoyancy forcing and their effects on the thermohaline circulation. This is also an important aspect of the ocean circulation and occurs over much longer timescales.

Page 3:

L 26:

"corresponding to positive [and neutral] Lyapunov exponents"

Page 4:

L 27 -28

In general, I don't like the use of the term 'this paper' in technical writing. First, it is incorrect - this is an online journal so this work will primarily be consumed as an electronic file. Second, it feels as if it is organized for the benefit of the writer, rather than the reader. Perhaps instead you could give the reader more context as to what they are about to read.

Instead of:

"1.3 This paper In this paper we wish..."

try,

"1.3 Programmatic Goals We wish to provide some first steps..."

Page 5:

L 1-2:

"In the present manuscript, we explore for the first time the Lyapunov spectra of a primitive-equation model, PUMA, and of the intermediate-order coupled oceanatmosphere system, MAOOAM."

I believe the Lyapunov spectra of MAOOAM was already studied to some degree in Vannitsem and Lucarini (2016), the wording here makes it sound like the authors are claiming it is explored here for the first time. Perhaps reword, for example:

"In the present manuscript, we explore the Lyapunov spectra of the intermediate-order coupled ocean-atmosphere system MAOOAM, and for the first time, of the primitive-equation model PUMA."

Page 6:

L 1:

For consistency, I suggest to change the order of the listed prognostic variables to match the order presented in the equations 1,2,3,4 below.

Page 8:

Table 1: Typo: "surface pressure pressure"

Page 10:

L 1:

"in a synthetic form"

What is synthetic about this? Perhaps you could just say,

"as a dynamical system"

L 25:

"2. Every time step, the model propagator is computed from the tangent linear model. This is the matrix that quantifies the transition from one model state into that one time step later."

This could be worded more clearly. Please clarify the definitions of the resolvent matrix, model propagator, and tangent linear model, and make sure to use the terms consistently for the remainder of the text.

"3. The model is integrated forward in time, and the propagators are accumulated (multiplied) into a matrix P"

It seems the more general procedure would be to integrate the linear propagator (e.g. using a geometric integrator / Magnus Expansion), but that this 'accumulation' via multiplication serves as an approximation. Perhaps you could be more precise about this statement.

Page 11:

L 5-6:

It seems odd to me that you cite a different author than Kaplan and Yorke for the Kaplan-Yorke dimension.

Kaplan, J. L. and Yorke, J. A. In Functional Differential Equations and Approximations of Fixed Points: Proceedings, Bonn, July 1978 (Ed. H.-O. Peitgen and H.-O. Walther). Berlin: Springer-Verlag, p. 204, 1979.

L 18:

"Finite-time Lyapunov exponents (FTLEs)"

You have already used this acronym before defining it here.

" (e.g. Haller, 2000)"

Perhaps you should instead cite one of the originators of the idea of FTLEs, e.g.

Abarbanel, H. D. I., R. Brown, and M. B. Kennel, "Variation of Lyapunov Exponents on a Strange Attractor," Journal of Nonlinear Science, 1, 175–199 (1991).

L 22:

"If a dynamical system is an Axiom A system or –invoking the chaotic hypothesis – one of a certain type of non Axiom A systems, these fluctuations for a finite, but large M may be described (based on (Schalge et al., 2013; Pazó et al., 2013; Laffargue et al., 2013)) by a large deviation law (Kifer, 1990; Touchette, 2009)."

This sentence is a bit clumsy. Perhaps you could reword or break into two sentences to make it easier to read.

L 29:

Make the definition of I() on its own line and given an equation number.

Page 13:

I'm not sure that I understand the table caption: "Common parameter values for the different model configurations of MAOOAM."

There is only one value given for each parameter.

Do you mean, "Model parameter values that are identical across all MAOOAM configurations used in this study"?

Page 16:

L 6:

"consequence of the non-existing clear-cut time-scale separation"

Please find another way to say this.

—

It would be nice if you could elaborate somewhere how you define the 'timescale' and units of the Lyapunov exponents, how you expect that to influence the prediction range, and explicitly how you expect these scales to map to different spatial scale instabilities. It seems to be mentioned in passing in a few places, but it would be helpful to summarize in one place before going into the results.

Page 18:

"The highly populated central manifold of MAOOAM is in stark contrast with the few near-zero LEs in PUMA. Being a purely atmospheric model, PUMA's Lyapunov spectrum does not exhibit the large time-scale separation present in MAOOAM. Indeed, the spectrum of PUMA bears more resemblance to that of the QG two-layer model of Schubert (2015)."

I'm curious if the authors have run their MAOOAM model in a forced-atmosphere and forced-ocean mode and computed LEs in order to demonstrate that the central manifold is largely eliminated without active coupling?

L 9:

"The additional positive and near-zero exponents that are introduced at these scales nevertheless indicate that the added resolution still resolves some scales that are important for the description of the dynamics. "

This implies that the number of positive LEs should asymptote as the resolution reaches a level to capture all relevant scales. Is this the expectation?

Page 19:

L 4:

"for the models [that] do not include"

Page 20:

L 8:

"The experiments [that] take this"

Page 25:

L 3-8:

I think this is an incredibly important passage, and should be investigated further to guide the development of coupled atmos/ocean systems.

Page 28:

"The source code to compute the Lyapunov exponents is available upon request to the corresponding author."

Please either include it as part of the supplemental material or make is available, for example, as part of the package: http://github.com/Climdyn/MAOOAM

---

## Author Comment (AC1) · 23 Apr 2018

**Author's Response to Anonymous Referee #1's Comments on "Exploring the Lyapunov instability properties of high-dimensional atmospheric and climate models"**

Lesley De Cruz[1], Sebastian Schubert[2], Jonathan Demaeyer[1], Valerio Lucarini[2,3,4], and Stéphane Vannitsem[1]

[1]Royal Meteorological Institute of Belgium, Brussels, Belgium
[2]Meteorological Institute, CEN, University Of Hamburg, Germany
[3]Department of Mathematics and Statistics, University of Reading, UK
[4]Centre for the Mathematics of Planet Earth, University of Reading, UK

*Correspondence to:* Lesley De Cruz (lesley.decruz@meteo.be)

The authors thank the referee for their careful reading of the manuscript and for providing helpful recommendations and pertinent remarks. Please find our response to each of your questions or remarks below.

This manuscript presents a Lyapunov analysis of two models: PUMA (purely atmospheric) and MAOOAM (atmosphere-ocean coupled). The purpose of this is to investigate the impact of different configurations (resolution, dissipation, and atmosphere-ocean coupling) on the instabilities of the system. This is an interesting path of research since, ultimately, it will permit to understand better the goodness of a certain model for forecasting certain processes operating at particular spatio-temporal scales.

While I consider the topic of this work is worth of being published, and the manuscript reasonably well written, I have one important criticism on the methodological procedure that prevents me from recommending its publication.

An important part of the manuscript is devoted to the rate function of the finite-time Lyapunov exponent (FTLE) distribution. As claimed in the abstract

1) For the PUMA model: "The convergence rate of the rate function(al) for the large deviation law of the FTLEs is fast for all exponents".

2) For the MAOOAM: "[...] it is possible to robustly define large deviation laws describing the statistics of the FTLEs corresponding to the strongly damped modes, [...]"

My main criticism is the meaningfulness of the rate function analysis considering the data used and their quality. I itemize next my concerns (a-c):

a) In all cases the claimed convergence is far from apparent with the naked eye. In my view the rate functions vary consistently as parameter tave changes, but I don't see a true convergence of the data as such. I find very questionable sentences like "For all LEs the tendency for convergence of the rate function is visible" in page 16, line 3; and "...FTLES accurately obey large deviation laws ..." in page 27. To judge the convergence from the shift of a whole curve as a parameter changes is really problematic.

Indeed, the graphs shown in the manuscript do not unambiguously prove the convergence of the rate functions. We have therefore moderated these statements as follows. Before, the relevant passage on page 16 read:

"We make the following observations. For all LEs the tendency for convergence of the rate function is visible. Also, the rate functions' shape is approximately parabolic and the estimates of the rate functions converge to the asymptotic with a comparable speed regardless of the value of the corresponding LE. "

This has been adapted as follows:

"We make the following observations. **The graphs suggest a convergence of the rate function for all LEs.** Also, the rate functions' shape is approximately parabolic and the estimates of the rate functions **appear to** converge to the asymptotic with a comparable speed regardless of the value of the corresponding LE. "

The following sentence on page 25, line 10:

" Similarly to what was found in a previous analysis performed on a severely truncated version of MAOOAM (Vannitsem and Lucarini, 2016), we find that the time series of the FTLEs corresponding to the strongly damped mode are weakly correlated and one can construct the rate functions defining the large deviations laws; compare figures 19a-c) for the 351 st LE."

has been corrected and now reads:

"Similarly to what was found in a previous analysis performed on a severely truncated version of MAOOAM (Vannitsem and Lucarini, 2016), we find that the time series of the FTLEs corresponding to the strongly damped modes are weakly correlated. **This would suggest that one can construct the rate functions defining the large deviations laws. The rate functions are shown in Fig. 20a-c) for the 351st LE. Their convergence properties are investigated in Appendix A2, and indicate that we have not yet converged to the central limit theorem even for these strongly damped modes.**

The sentence on page 27, line 7:

" [. . . ] one finds that all the FTLEs accurately obey large deviation laws [. . . ]"

has been adapted and now reads:

" [. . . ] **the results suggest** that the FTLEs obey large deviation laws [. . . ]"

Furthermore, the paragraph on page 27, lines 24-27:

"The analysis of the FTLEs of MAOOAM reveals some interesting insight into the dynamics. The FTLEs associated to the strongly dissipative modes obey large deviation laws, while those corresponding to the near-zero LEs do not. This behaviour is expected, and in agreement with what was found in Vannitsem and Lucarini (2016). Surprisingly, however, it is hard to find convergence for the FTLEs associated to the positive LEs. This may point to the presence of nontrivial ocean influence on the (mostly) atmospheric instabilities."

now reads:

"The analysis of the FTLEs of MAOOAM reveals some interesting insight into the dynamics. Surprisingly, **it is hard to find convergence for the rate functions of the FTLEs, even for those associated to the positive LEs**. This may point to the presence of nontrivial ocean influence on the (mostly) atmospheric instabilities."

Finally, the sentence from the abstract:

"In all considered configurations, it is possible to robustly define large deviations laws describing the statistics of the FTLEs corresponding to the strongly damped modes, while the opposite holds for near-zero LEs and, somewhat unexpectedly, also for the positive LEs."

has been adapted and now reads:

"In all considered configurations, **we are not yet in the regime in which one can robustly define large deviations laws describing the statistics of the FTLEs**."

b) The data analysis resorted to a strong smoothing of the data obtained from short time series, cf. the histograms. It is difficult to evaluate the errors accumulated thereby.

Smoothing or binning the data is an unavoidable step in order to evaluate the PDF. For this we have employed the kernel density estimation method of Scott (1979; see below) which is optimised to rapidly converge to the true underlying distribution.

c) From a theoretical point of view, I have doubts that the rate function can be detected with the time intervals over which the FTLEs are computed ("tave"). I'm afraid that the values of "tave" used are simply too small to reach the asymptotic rate function, even if the time series were infinitely long. From a theoretical perspective the choice of a reference time "T", based on the time interval in which autocorrelations of the FTLE decay below 1/e, presents certain problems. The rate function is calculated for times only up to 28*T, what may be too small to detect the rate function (independently of the amount of data). I say this because in many spatio-temporal chaotic systems the "renewal time" of the Lyapunov vector operates at a time scale T_x much larger than T.

In such a case, in order to detect the rate function one must take time intervals (tave) larger than T_x (> > T), see Pazo et al (2013). For instance, in Laffarge et al (2013), with 40 coupled maps the time interval (tave) used to measure the rate function is 10^4 iterations.

The only way I see to demonstrate something unambiguously with the numerical data at hand is to check the convergence of a particular quantity (instead of a curve). Following Pazo et al (2013), and as double-check, I suggest to measure the variance of the FTLE for different "tave" values. Multiplying by tave, it should be possible to verify if the data level off at a certain value D in the range of tave values considered. (The diffusion coefficient D is the inverse of the second derivative of the rate function at its minimum, see e.g. (Kuptsov and Politi, 2011)).

As suggested, we have computed $\sigma$ for a range of different averaging block lengths $t_{ave}$ to verify the expected scaling behaviour. The results are shown in Fig. 1 for the PUMA model. Furthermore, we have compared the value of $\sigma^2 \cdot t_{ave}$ to the diffusion coefficient $D$. The results for PUMA are shown in Fig. 2, and suggest that the value of $\sigma^2 \cdot t_{ave}$ appear to converge. While the value of $D$ fluctuates, it has the right order of magnitude.

The corresponding graphs for MAOOAM are shown in Figs. 3 and 4. Both the values of $D$ and $\sigma^2 \cdot t_{ave}$ vary as a function of $t_{ave}$, indicating that much longer integration times are required than the 30 years used here to investigate the rate function. A discrepancy between $D$ and $\sigma^2 \cdot t_{ave}$ is apparent for LE 100 in all experiments shown in Fig. 4. This can be explained by the extremely long decorrelation times for these near-zero LEs, due to the multiscale properties of the system.

These results have been added to the manuscript as an Appendix.

[Figure]

**Figure 1.** Standard deviation $\sigma$ as a function of the block averaging length $t_{ave}$ for different Lyapunov exponents of the PUMA model. The Lyapunov index is shown in the legend. The top panel shows the results for a temperature gradient $\Delta T_{EP}$ of 50K, the bottom panel for 60K. The black dashed line corresponds to $t_{ave}^{-\frac{1}{2}}$ scaling.

[Figure]

**Figure 2.** The metric $\sigma^2 \cdot t_{ave}$ versus the diffusion coefficient $D$, derived from the inverse of the second derivative at the minimum of the rate function, as a function of the block averaging length $t_{ave}$ for different Lyapunov exponents of the PUMA model. The Lyapunov index is shown in the title. The top panels show the results for a temperature gradient of 50K, the bottom panels for 60K.

Minor comments: 1. Eq. (19), please mention that M^* is the adjoint of M. Note also that M has a wrong font type in Eq. (19).

We have corrected the font and added the following sentence after Eq. (19):

"where $\mathbf{M}^*$ is the adjoint of $\mathbf{M}$."

2. Letter M is used for the resolvent matrix and for the integer (tave= T*M). I would avoid this duplicity.

We have replaced the letter $M$ by the lowercase letter $m$.

3. The concept backward Lyapunov exponent appears in page 10, line 22, without much explanation. Note that Lyapunov exponents obtained from Eq. (19) are actually "forward Lyapunov exponents". The mirror definition of Osedelets theorem with M M^*, instead of M^* M, yields backward LEs. I point to table 1 in Pazo et al (2010) and to (Ershov & Potapov, 1998. On the concept of stationary Lyapunov basis. Physica D 118(3-4), 167–198.) for the formal link between Osedelets theorem and Bennetin's algorithm.

Indeed, the algorithm we have used produces the backward Lyapunov exponents. The order of the matrices in Eq. (19) has been corrected and now reads $\mathbf{MM}^*$. We have also adapted the sentence:

"The *Lyapunov exponents* are then defined as the natural logarithm of the eigenvalues of $\Lambda_{x_0}$."

This sentence now reads:

"The **backward** *Lyapunov exponents* (**Ershov and Potapov, 1998; Pazó et al., 2010**) are then defined as the natural logarithm of the eigenvalues of $\Lambda_{x_0}$."

[Figure]

**Figure 3.** Standard deviation $\sigma$ as a function of the block averaging length $t_{ave}$ for different Lyapunov exponents of the 9x9 configuration of the MAOOAM model. The Lyapunov index is shown in the legend. The top left panel shows the results for the experiment without scale-dependent dissipation, the top right panel corresponds to the reference value for dissipation and the lower panel shows the enhanced dissipation results. The black dashed line corresponds to $t_{ave}^{-\frac{1}{2}}$ scaling.

4. Page 11. The relationship of the KY dimension with the fractal dimension was confusing for me as written now (probably due to the intention of making it simple for the unfamiliar reader). The KY dimension is an estimation of the information dimension, usually denoted D_1. D_1 is known to be (equal or) smaller than the capacity or box-counting dimension (D_0), which I guess is what the authors refer to by "fractal dimension", following the paper by Frederickson et al (1983). My taste is that nowadays one can talk of D_KY as estimation of D_1, and perhaps to mention the information dimension bounds the capacity/fractal dimension (for the unfamiliar reader on these questions).

Thank you for pointing out this ambiguity. The original version read:

"[. . . ] $D_{KY}$, which provides (a lower bound on) the fractal dimension of the attractor, and is defined as (Frederickson et al., 1983): [. . . ]"

We have adapted this following your suggestion:

[Figure]

**Figure 4.** The metric $\sigma^2 \cdot t_{ave}$ versus the diffusion coefficient $D$, derived from the inverse of the second derivative at the minimum of the rate function, as a function of the block averaging length $t_{ave}$ for different Lyapunov exponents of the 9x9 configuration of the MAOOAM model. The Lyapunov index is shown in the title. The top panels show the results for the experiment without scale-dependent dissipation, the centre panels correspond to the reference value for dissipation and the lower panels show the results for an enhanced dissipation coefficient.

"[...] $D_{KY}$, which **is an estimation of the information dimension $D_1$. $D_1$ is known to be less than or equal to the capacity or box-counting dimension $D_0$ , also referred to as the fractal dimension (Frederickson et al., 1983).** $D_{KY}$ is defined as **(Kaplan and Yorke, 1979)**: [...]"

5. Page 11, when introducing the FTLEs the authors refer to Haller (2000). I have nothing against, but I think it is more appropriate H. Fujisaka, Prog. Theor. Phys. 70, 1264 (1983)

Indeed. We have replaced the reference to Haller by the references suggested by you and the other referee:

- H. Fujisaka, Progress of Theoretical Physics 70, 1264 (1983)

- H.D. Abarbanel et al., Journal of Nonlinear Science 1, 175–199 (1991)

6. Page 15, line 1, I think the use of "much smaller" is exaggerating the difference between the spectra. Using "smaller" is enough.

Indeed, we have corrected the wording accordingly.

7. Page 15, line 4, when mentioning the Tibaldi-Molteni index, why is not the paper of Tibaldi and Molteni (1990) cited instead?

Thank you for pointing this out. The reference has been adapted accordingly.

8. It should be said somewhere that Fig. 1 shows (only) the 200 largest LEs.

We have adapted the sentence by adding the text in boldface:

"Figure 1 shows **the 200 largest LEs of** the two different Lyapunov spectra obtained in our experiments with PUMA."

The caption of Fig. 1 has been adapted accordingly:

"The **200 largest LEs of the** Lyapunov spectra of PUMA for the two different setups with $\Delta T_{EP} = 50K$ and $60K$."

9. In Fig. 1, it looks like the Lyapunov index starts at 0, instead of 1. I guess the lines have to be displaced 1 unit rightwards.

We have recreated the graph with the index starting at 1 instead of 0.

10. Page 15, line 9, "faster" -> "fast" (?)

Indeed, "faster decaying LE 150" is now replaced by "**fast-decaying** LE 150".

11. In Fig. 2 it is not said which kernel function is used, or how the bandwidth was optimized.

The kernel function used was the one by Scott, D., "On optimal and data-based histograms", Biometrika 66 (3): 605–610, doi:10.1093/biomet/66.3.605 (1979). We have added this reference and adapted the sentence:

"The top panels of these figures show the approximation of the respective distributions obtained via kernel smoothing of the distribution of the block-averaged LEs. "

It now reads:

"The top panels of these figures show the approximation of the respective distributions obtained via **kernel density estimation** (Scott, 1979) of the distribution of the block-averaged LEs. "

12. Figs. 4,5 the x-tic labels overlap.

Thank you for pointing this out, this has been corrected.

[Figure]

**Figure 5.** Lyapunov spectra for the different MAOOAM experiments as a function of the *rescaled* Lyapunov index $i/N$.

13. Figs. 11-14. I'm curious how the Lyapunov spectra look like when the x-coordinate (the Lyapunov index) is rescaled by the number of degrees of freedom. Is there some overlapping of the data for the most negative LEs?

The Lyapunov spectra for MAOOAM are plotted as a function of the rescaled Lyapunov index $\lambda_i/N$ in Fig. 5. The different regions do seem to overlap. The values of the most negative LEs only overlap in the experiments with *scale-independent* dissipation ("nodissip").

**Author's Response to S.G. Penny's Comments on "Exploring the Lyapunov instability properties of high-dimensional atmospheric and climate models"**

Lesley De Cruz[1], Sebastian Schubert[2], Jonathan Demaeyer[1], Valerio Lucarini[2,3,4], and Stéphane Vannitsem[1]

[1]Royal Meteorological Institute of Belgium, Brussels, Belgium
[2]Meteorological Institute, CEN, University Of Hamburg, Germany
[3]Department of Mathematics and Statistics, University of Reading, UK
[4]Centre for the Mathematics of Planet Earth, University of Reading, UK

*Correspondence to:* Lesley De Cruz (lesley.decruz@meteo.be)

The authors thank S.G. Penny for the thorough reading of the manuscript, the supportive comments and the constructive remarks. The manuscript has benefited a lot from your helpful input.

Below is a point-by-point list of modifications that have been applied, based on your report.

General points:

In general, this is a very nicely written paper. The introduction is accessible and informative. The results are interesting and I believe will inspire a number of new research directions.

I'd suggest that the authors perform a forced atmosphere-only and a forced ocean-only experiment with the MAOOAM system to compare how typical forced atmosphere or ocean models compare to coupled systems in terms of the Lyapunov spectrum. Or if it already exists, point to a previous work by the authors that has done this comparison. This would be helpful for guidance to the operational centres currently making decisions about what is gained from transitioning from separate component forecast systems to a fully coupled forecast system.

As a general comment, the authors should strive to cite the original works for various concepts rather than a reference text or review paper.

To the editor: Regarding the journal's typesetting decisions, please place the figures closer to where they are mentioned in the text.

Technical points:

L 9-10:

"but also the errors that are present either in the model parametrizations, known as model errors,"

It would be more accurate to say this is known as 'model parameterization error', since 'model error' includes systematic misrepresentation of the system dynamics.

Indeed, the term model error is too general here. We have removed the inaccurate part of the sentence, which now reads: "This sensitivity property affects not only the dynamics of errors in the initial conditions but also the errors that are present either in the model parametrizations or in the boundary conditions [...]."

L 11:

Missing period at the end of the line.

Corrected.

L 30:

This paragraph should start with a "However,"

We agree; this has been added to the paragraph.

L 32:

" for atmospheric instabilities, and most notably convective"

Remove the word 'and'

The superfluous 'and' has been removed.

L 34:

"The oceanic circulation, by contrast, is mostly mechanically driven by atmospheric winds"

This is true on shorter timescales, but you should also acknowledge buoyancy forcing and their effects on the thermohaline circulation. This is also an important aspect of the ocean circulation and occurs over much longer timescales.

Indeed, at longer timescales, buoyancy forcing is an important aspect on the ocean circulation that must be mentioned. The mechanisms of forcing of the deep oceanic circulation are also related to the winds, that favour the mixing. Other important factors are tides and wave breaking. There are various views on this by oceanographers.

The sentence has been adapted from the original version:

"The oceanic circulation, by contrast, is mostly mechanically driven by atmospheric winds [...]."

and now reads:

"The **surface** oceanic circulation, by contrast, is mostly mechanically driven by atmospheric winds [...]. **On even longer timescales, buoyancy fluxes are an important driver for the deep ocean's thermohaline circulation.**"

Page 3:

L 26:

"corresponding to positive [and neutral] Lyapunov exponents"

The sentence has been adapted as suggested. Indeed, the null exponents also form a part of the unstable subspace as used by Trevisan et al. (2010).

Page 4:

L 27 -28

In general, I don't like the use of the term 'this paper' in technical writing. First, it is incorrect - this is an online journal so this work will primarily be consumed as an electronic file. Second, it feels as if it is organized for the benefit of the writer, rather than the reader. Perhaps instead you could give the reader more context as to what they are about to read.

"1.3 This paper In this paper we wish..."

try,

"1.3 Programmatic Goals

We wish to provide some first steps..."

Thank you for pointing this out. We have implemented the changes you have suggested.

Page 5:

L 1-2:

"In the present manuscript, we explore for the first time the Lyapunov spectra of a primitive-equation model, PUMA, and of the intermediate-order coupled ocean- atmosphere system, MAOOAM."

I believe the Lyapunov spectra of MAOOAM was already studied to some degree in Vannitsem and Lucarini (2016), the wording here makes it sound like the authors are claiming it is explored here for the first time. Perhaps reword, for example:

"In the present manuscript, we explore the Lyapunov spectra of the intermediate-order coupled ocean-atmosphere system MAOOAM, and for the first time, of the primitive-equation model PUMA."

The Lyapunov spectra of MAOOAM (more precisely of its predecessor, VDDG; see Vannitsem et al. 2015) have indeed been studied before, but only for the low-order model configuration of 36 variables. It is, however, the first Lyapunov analysis of MAOOAM model configurations with hundreds of variables. We have changed the sentence as follows to clarify this:

"In the present manuscript, we explore for the first time the Lyapunov spectra of **the** primitive-equation model PUMA, and of **intermediate-order configurations of** the coupled ocean-atmosphere system, MAOOAM."

Page 6:

L 1:

For consistency, I suggest to change the order of the listed prognostic variables to match the order presented in the equations 1,2,3,4 below.

The sentence has been adapted to match the order of the equations: "The prognostic equations as written in PUMA's code have four prognostic fields, the relative vorticity $\eta$, the divergence $D$, **the logarithm of the surface pressure** $\ln p_s$ **and the temperature** $T$."

Page 8:

Table 1: Typo: "surface pressure pressure"

This has been corrected.

Page 10:

L 1:

"in a synthetic form"

What is synthetic about this? Perhaps you could just say,

"as a dynamical system"

The sentence has been adapted as suggested.

L 25:

"2. Every time step, the model propagator is computed from the tangent linear model. This is the matrix that quantifies the transition from one model state into that one time step later."

This could be worded more clearly. Please clarify the definitions of the resolvent matrix, model propagator, and tangent linear model, and make sure to use the terms consistently for the remainder of the text.

Indeed, this was worded ambiguously. We have adapted the text as follows. On P 10 L 13, have added the words in bold: "[...] the matrix $\mathbf{M}$ is referred to as the resolvent matrix **or propagator**."

Furthermore, we have clarified step 2 of the algorithm as follows:

"2. At every time step $t_i$, a matrix $\mathbf{P}_i$ that represents the linear propagator from $t_{i-1}$ to $t_i$ is computed using the tangent linear model along the model state trajectory. $\mathbf{P}_i$ is the equivalent of the matrix $\mathbf{M}$ for a finite time difference $t_i - t_{i-1}$. We take into account the numerical integration scheme when computing $\mathbf{P}_i$, by evaluating the model Jacobian at all intermediate points of the scheme. We have implemented the second- and fourth-order Runge-Kutta schemes, which require two and four evaluations of the Jacobian per time step, respectively."

"3. The model is integrated forward in time, and the propagators are accumulated (multiplied) into a matrix P"

It seems the more general procedure would be to integrate the linear propagator (e.g. using a geometric integrator / Magnus Expansion), but that this 'accumulation' via multiplication serves as an approximation. Perhaps you could be more precise about this statement.

We have changed the description of step 3 as follows:

"3. As the model is integrated forward from time $t_i$ to $t_{i+b}$, the corresponding linear propagator $\mathbf{P}_{i,i+b}$ is approximated by multiplying the $b$ matrices, $\mathbf{P}_{i,i+b} = \mathbf{P}_{i+b} \ldots \mathbf{P}_{i+1}$. In the experiments that follow, we have chosen $b = 1$."

Page 11:

L 5-6:

It seems odd to me that you cite a different author than Kaplan and Yorke for the Kaplan-Yorke dimension.

Kaplan, J. L. and Yorke, J. A. In Functional Differential Equations and Approximations of Fixed Points: Proceedings, Bonn, July 1978 (Ed. H.-O. Peitgen and H.-O. Walther). Berlin: Springer-Verlag, p. 204, 1979.

Thank you for pointing this out. This reference has been corrected.

L 18: "Finite-time Lyapunov exponents (FTLEs)" You have already used this acronym before defining it here.

We have moved the definition to the first occurrence of the acronym.

" (e.g. Haller, 2000)" Perhaps you should instead cite one of the originators of the idea of FTLEs, e.g. Abarbanel, H. D. I., R. Brown, and M. B. Kennel, "Variation of Lyapunov Exponents on a Strange Attractor," Journal of Nonlinear Science, 1, 175–199 (1991).

Indeed. We have replaced the reference to Haller by the references suggested by you and the other referee:

  – H. Fujisaka, Progress of Theoretical Physics 70, 1264 (1983)

  – H.D. Abarbanel et al., Journal of Nonlinear Science 1, 175–199 (1991)

"If a dynamical system is an Axiom A system or –invoking the chaotic hypothesis – one of a certain type of non Axiom A systems, these fluctuations for a finite, but large M may be described (based on (Schalge et al., 2013; Pazó et al., 2013; Laffargue et al., 2013)) by a large deviation law (Kifer, 1990; Touchette, 2009)." This sentence is a bit clumsy. Perhaps you could reword or break into two sentences to make it easier to read.

Indeed, this sentence is hard to read. We have rewritten it as follows:

"As discussed in (Schalge et al., 2013; Pazó et al., 2013; Laffargue et al., 2013), some dynamical systems have the property that for a finite, but large $M$, the fluctuations of their FTLEs can be described by a large deviation law (Kifer, 1990; Touchette, 2009). This is the case for Axiom A systems, and invoking the chaotic hypothesis, extends to certain types of non Axiom A systems."

L 29: Make the definition of I() on its own line and given an equation number.

We have changed this as suggested.

Page 13: I'm not sure that I understand the table caption: "Common parameter values for the different model configurations of MAOOAM." There is only one value given for each parameter. Do you mean, "Model parameter values that are identical across all MAOOAM configurations used in this study"?

Indeed, this is what we meant. We have changed the table caption accordingly.

L 6: "consequence of the non-existing clear-cut time-scale separation" Please find another way to say this.

We have adapted the original sentence:

"We interpret these results as another consequence of the non-existing clear-cut time-scale separation in a purely atmospheric model like PUMA."

as follows:

"**We interpret these results to stem from the lack of a** clear-cut time-scale separation in a purely atmospheric model like PUMA."

It would be nice if you could elaborate somewhere how you define the 'timescale' and units of the Lyapunov exponents, how you expect that to influence the prediction range, and explicitly how you expect these scales to map to different spatial scale instabilities. It seems to be mentioned in passing in a few places, but it would be helpful to summarize in one place before going into the results.

Thank you for this remark. We have added the following paragraph, just after the definition of the Lyapunov exponents:

"If one or more LEs are positive, small errors on the initial conditions of the system grow exponentially and the system is chaotic. In that case, the time horizon of the system's predictability is proportional to the inverse of the largest Lyapunov exponent, $\frac{1}{\lambda_1}$. As this predictability horizon is expressed in days for operational forecasting, we also express the exponents $\lambda_i$ in units $day^{-1}$. To translate the spectrum of LEs into spatial scales of the instabilities in an unambiguous way, the CLVs must also be determined. However, if there is scale-dependent dissipation, the largest negative LEs are likely to be associated with the smallest, most dissipative scales."

[Figure]

**Figure 1.** Lyapunov spectra of MAOOAM without ocean dynamics or in "slab ocean" mode, for model configurations from atm. 2x4, oc. 2x4 (red full line) up to atm. 8x8, oc. 8x8 (brown dashed line). Lyapunov exponents are ranked in decreasing order, and the index of the smallest positive Lyapunov exponent is indicated with a downward-pointing arrow for each model configuration.

"The highly populated central manifold of MAOOAM is in stark contrast with the few near-zero LEs in PUMA. Being a purely atmospheric model, PUMA's Lyapunov spectrum does not exhibit the large time-scale separation present in MAOOAM. Indeed, the spectrum of PUMA bears more resemblance to that of the QG two-layer model of Schubert (2015)."

I'm curious if the authors have run their MAOOAM model in a forced-atmosphere and forced-ocean mode and computed LEs in order to demonstrate that the central manifold is largely eliminated without active coupling?

The Lyapunov spectra of forced-atmosphere models such as those of Charney and Straus (1970) and Marshall and Molteni (1993) have been computed, for example, in: Vannitsem, "Predictability of large-scale atmospheric motions: Lyapunov exponents and error dynamics", Chaos 27, 032101 (2017). In this case, the spectrum does not display the large set of near-zero exponents.

We have performed some additional simulations without ocean dynamics, the results of which are shown in Fig. 1. The structure of the spectrum is similar to the one presented in the manuscript, and is associated with the exchange of energy between the ocean (a thermal bath) and the atmosphere.

No ocean-only experiments have been done with MAOOAM because it would imply that (i) temperature will no longer play a role in the dynamics (since it is just a passive scalar) and (ii) one must impose some specific wind stress forcing. This question was investigated using the low-order coupled ocean-atmosphere model OA-QG-WS v2, a predecessor of MAOOAM, in Vannitsem, "Stochastic modelling and predictability: analysis of a low-order coupled ocean–atmosphere model", Phil. Trans. R. Soc. A 372.2018, 20130282 (2014). There, the ocean system was found to converge towards a constant value when a constant surface forcing, consistent with the fully coupled dynamics, is applied.

L 9:

"The additional positive and near-zero exponents that are introduced at these scales nevertheless indicate that the added resolution still resolves some scales that are important for the description of the dynamics. "

This implies that the number of positive LEs should asymptote as the resolution reaches a level to capture all relevant scales. Is this the expectation?

Yes, this is expected when the resolution suffices to capture all relevant dynamics.

Page 19:

L 4:

"for the models [that] do not include"

Fixed.

Page 20:

L 8:

"The experiments [that] take this"

Fixed.

Page 25:

L 3-8:

I think this is an incredibly important passage, and should be investigated further to guide the development of coupled atmos/ocean systems.

This is indeed an important and counterintuitive result, especially since forced-ocean models appear to be quite stable under a constant forcing. We have added Fig. 2 to the manuscript, and have added the sentence:

"Indeed, the quantity $D_{KY}/N$, which approximates the relative fraction of the attractor's dimension, increases for increasing ocean resolution, but decreases for increasing atmosphere resolution, as illustrated in Fig. [2]. "

Page 28:

"The source code to compute the Lyapunov exponents is available upon request to the corresponding author."

Please either include it as part of the supplemental material or make is available, for example, as part of the package: http://github.com/Climdyn/MAOOAM

We have created a branch in the MAOOAM git repository and tagged the version **v1.3.1-lyapunov** that was used to perform the simulations in this manuscript. This version is archived on Zenodo (https://doi.org/10.5281/zenodo.1198650).

The code availability section concerning MAOOAM has been adjusted to:

[revised manuscript text omitted]

---

## Editor Comment (EC1) · JM Lopez (Editor) · 28 Apr 2018

In my opinion, all referee questions have been satisfactorily addressed by the authors response and the new version of the manuscript. Therefore, I encourage the authors to submit a final version of the paper for publication including the changes made as they are in the current resubmission.